# Bioactive lipid-mediated structural and functional regulation of the essential human potassium channel Kir7.1

Qingwei Niu [1,2,3,4,9], Simon Vu[2,4,9], Yuanjian Xu [5], Mingxing Qian [5], Anastasiia Rudenko[1,2,6], Jin Ye [4], Jianwei Zeng [4], Wei Huang [7], Douglas F. Covey [5], Rui Zhang [4] ✉, Ziao Fu [1,2] ✉ & Polina V. Lishko [1,2,8] ✉

The inwardly rectifying potassium channel Kir7.1 is essential for the physiological function of diverse tissues, including the retinal pigment epithelium and the gestational myometrium. Loss-of-function mutations in *KCNJ13*, which encodes Kir7.1, or conditional ablation of Kir7.1 in the retinal pigment epithelium, lead to early-onset vision loss. Despite strong genetic evidence supporting Kir7.1 as a therapeutic target, its regulation by endogenous ligands—beyond phosphoinositides—remains poorly understood. Here, we report cryo-electron microscopy structures of human Kir7.1 in multiple functional states at resolutions ranging from 2.8 Å to 4.0 Å. These structures uncover the molecular basis of Kir7.1 modulation by $PI_{4,5}P_2$, its selectivity, rectification, and identify a distinct steroid-binding site that may mediate cooperative channel gating. Our data suggest that endogenous cholesterol acts as an inhibitory ligand, which is displaced by select activating steroids. These activating steroids work in concert with $PI_{4,5}P_2$ to promote channel opening through profound changes in cytoplasmic domains, and the linker region. Electrophysiological analyses define a pharmacological landscape of Kir7.1 activators, providing innovative tools to probe and modulate channel function in both physiological and pathological contexts.

The inwardly rectifying potassium channel Kir7.1[1] (encoded by *KCNJ13*) is critical for normal functioning of the retinal pigment epithelium (RPE), an essential tissue that supports photoreceptors by maintaining the ionic homeostasis of the subretinal space[2,3] and waste removal[1,4–8]. Loss-of-function mutations in *KCNJ13* are associated with blindness in humans[5,6,9–11] and targeted deletion of Kir7.1

in RPE leads to progressive photoreceptor degradation in both in vivo and in vitro[10,12,13].

Kir7.1 is highly conserved evolutionarily, sharing 92.6% amino acid identity between humans and rodents, and belongs to the broader family of inwardly rectifying potassium (Kir) channels that regulate cellular excitability and epithelial ion transport across a wide range of

[1]Department of Cell Biology and Physiology, WashU Medicine; Washington University School of Medicine, St. Louis, MO, USA. [2]Center for Investigation of Membrane Excitability Diseases (CIMED), WashU Medicine; Washington University School of Medicine, St. Louis, MO, USA. [3]Molecular and Cell Biology/DBBS graduate program, WashU; Washington University in St. Louis, St. Louis, MO, USA. [4]Department of Biochemistry and Molecular Biophysics, WashU Medicine; Washington University School of Medicine, St. Louis, MO, USA. [5]Department of Developmental Biology and Taylor Family Institute for Innovative Psychiatric Research, WashU Medicine; Washington University School of Medicine, St. Louis, MO, USA. [6]Department of Cell Biology and Physiology Postbacc program, WashU Medicine; Washington University School of Medicine, St. Louis, MO, USA. [7]Department of Pharmacology, Case Western Reserve University, Cleveland, OH, USA. [8]BJC Investigator Program, WashU Medicine, St. Louis, MO, USA. [9]These authors contributed equally: Qingwei Niu, Simon Vu. ✉e-mail: zhangrui@wustl.edu; ziao@wustl.edu; lishko@wustl.edu

tissues[14]. Beyond its ocular function, Kir7.1 also contributes to the regulation of feeding behavior[15], and modulates uterine contractions during pregnancy, implicating it in the control of parturition[16,17].

Kir7.1 activity is known to be enhanced by phosphoinositide lipid $PI_{4,5}P_2$ ($PIP_2$)[8,18]. We previously identified two endogenous neurosteroids−progesterone (P4) and dehydroepiandrosterone (DHEA)−as the first two small molecule activators of Kir7.1[16,19], along with two synthetic activators: 17-α-hydroxyprogesterone caproate (17-OHPC) and dydrogesterone[16]. However, the absence of the Kir7.1 atomic structure has hindered our understanding of its functional mechanisms and activation by ligands.

In this work, we purified recombinant human Kir7.1 expressed in mammalian cells and determined cryo-EM structures at 2.8–4.0 Å resolution in multiple functional states, including $PIP_2$-bound extended "E"-state; $PIP_2$-bound docked "D"-state, and an agonist-bound state. This set of structures provides critical insights into the gating mechanisms of Kir7.1, revealing a cooperative interaction between phosphoinositides and a steroid binding site. Electrophysiological analyses further demonstrate strong synergy between $PIP_2$ and specific activating steroids. Notably, steroids such as progesterone can activate Kir7.1 even in the absence of $PIP_2$, but the presence of $PIP_2$ enhances steroid binding and significantly amplifies channel activation. Interestingly, endogenous cholesterol competes with activating steroids for binding, suggesting a dynamic regulatory mechanism. We further screen a panel of endogenous and synthetic steroids and their stereoisomers, refining the pharmacological profile of Kir7.1 modulators. Together, these studies not only provide detailed structural and functional insights into Kir7.1 regulation but also identify compounds that may serve as prototypes for therapeutic strategies aimed at restoring Kir7.1 function in disease-affected tissues.

## Results

### Purification and characterization of the human Kir7.1 atomic cryo-EM structure

To obtain sufficient quantities of high-quality protein for structural analysis, a human Kir7.1 construct (pKCNJ13-TEV-mEmerald-10 × His) was expressed in Expi293F™ GnTI- cells (Fig. S1a, b). Channel functionality was confirmed by whole-cell patch-clamp recordings, verifying that the appended tags did not interfere with Kir7.1 activity (Fig. S1c, d). For structural studies, the protein was purified using a two-step purification protocol comprising affinity chromatography followed by size-exclusion chromatography (Fig. S2a, b). The size-exclusion chromatography elution profile of the Kir7.1 (Fig. S2b) showed a prominent elution peak between 14 mL and 15 mL, corresponding to its tetrameric structure.

The structure of the human Kir7.1 channel was resolved by single-particle cryo-EM analysis (Figs. 1a–d, S2c, d, and S3 and Table 1) to an overall resolution of 3.3 Å providing initial structural insights into this physiologically essential potassium channel (Fig. 1a–d).

The local resolution map shows that significant portion of the transmembrane domain (TMD), particularly the selectivity filter, as well as parts of cytoplasmic domain (CTD) reached 2.8 Å resolution (Fig. S4a, d). The final structural model revealed all typical structural features found in Kir-type family of potassium channels, including the canonical selectivity filter and the gating machinery comprised of G-loop and the inner helix gates (Fig. 1e). Structural alignment of Kir7.1 with previously resolved structures of homologous Kir channels in closed "apo" conformations- Kir2.2[20] (PDB: 3JYC), Kir3.2[21] (PDB: 6XIS), and Kir6.2[22] (PDB: 6C3P)- revealed a conserved architecture of its selectivity filter, notably the GYG motif that is characteristic for potassium channel superfamily (Figs. 1f and S5). As shown on Fig. 1g, the cryo-EM density map of Kir7.1 displayed additional electron density within the pore, consistent with potassium ions coordinated at canonical binding sites.

A key structural distinction in Kir7.1 occurs at position 125, where a methionine (Met125) replaces the highly conserved arginine typically found in other Kir family members, e.g., Kir2.2, Kir3.2, and Kir6.2 (Figs. 1f−h and S5). This substitution likely underpins the ion selectivity of Kir7.1, particularly its permeability to cesium ions[1]- a trait uncommon among Kir or even other potassium channels, in which cesium usually acts as a pore blocker. Arginine is positively charged, forming a salt bridge with conserved glutamate (typically ~2.7 Å apart in all other Kir channels, (Fig. 1h), thus, helping reinforce narrow passage and an ion selectivity. In Kir7.1, this salt bridge is lost, due to nonpolar nature of Met125 bearing a flexible thioether side chain. This change results in a wider, 4.6 Å distance between Met125 and Glu115, and a more permissive conformation in the filter region. A weaker electrostatic constraints on ion passage, broadens ion selectivity and, thus, may allow $Cs^+$ permeation (Fig. 1h).

### Two conformational states of Kir7.1

Detailed analysis of the cryo-EM dataset using three-dimensional classification revealed two distinct conformational states of Kir7.1. One subset is referred to as the extended or E-state (Figs. 1, 2a, S3, and S4a, d and Table 1) while the second subset is designate as the docked or D-state (Figs. 2b, S4b, e, and S6 and Table 1). Structural comparison between the E- and D- states revealed substantial conformational rearrangements. Notably, the transition from the E- to the D-state brings the CTD approximately 10 Å toward to the membrane plane, resulting in a reduction of the overall channel length from 115 Å to 105 Å. Interestingly, comparison of the selectivity filter between two conformations revealed a slightly widening of the channel's outer mouth, increasing from 6.4 Å (E-state) to 8.6 Å (D-state), as measured by the carbonyl distance of Y122 across opposing subunits (Fig. S7a). The remaining residues within the ion selectivity filter retained similar backbone distances, indicating relative structural stability in this region (Figs. 2c and S7). However, we observed larger conformational changes in the outer helices of TMD which transformed from a relatively straight to an inwardly kinked shape (Fig. 2c). This rearrangement was the most prominent at a turning point located between Val65-Val65 from opposite monomers, resulting in 2.48 Å shift from 37.7 Å (E-state) to 35.2 Å (D-state; Figs. 2c and S7).

Even larger conformational rearrangements were observed in CTD, which revealed a pronounced 45° clockwise rotation of the CTD along a helical path in a screw-like motion (Fig. 2d and Supplementary Movie 1). This represents one of the largest conformational rearrangement reported to date among this family of potassium channels[21]. By comparing the monomer model between these two states, we found this CTD rotation is facilitated by the linker region connecting the TMD and the CTD (Fig. 2e). In the E-state, the linker stretches to a flexible loop, whereas in the D-state, it recoils into an α-helix (Fig. 2e). This secondary structure change contributes to the structural flexibility necessary for the channel to shift between extended and docked states in response to different functional environments suggesting a potential ligand exchange or a binding site occupancy.

### Kir7.1 associates with endogenous phosphatidylinositol 4,5-bisphosphate in D-state

Through model building and density fitting, we identified a prominent non-protein density at the base of the outer helix in the D-state of Kir7.1, corresponding to the conserved phosphatidylinositol 4,5-bisphosphate ($PIP_2$) binding site (Figs. 3a, b and S4j). This site aligns with the well-characterized $PIP_2$ interaction pocket conserved across the Kir channel family[21,23–25], which is known to play a central role in regulating channel opening (Fig. 3c). Interestingly, no synthetic or exogenous $PIP_2$ were added during protein purification process, therefore this density corresponds to the endogenous $PIP_2$ produced by the expression system itself, i.e., Expi293F™ GnTI- cells. Upon docking a $PIP_2$ molecule into the density, the phosphate headgroup

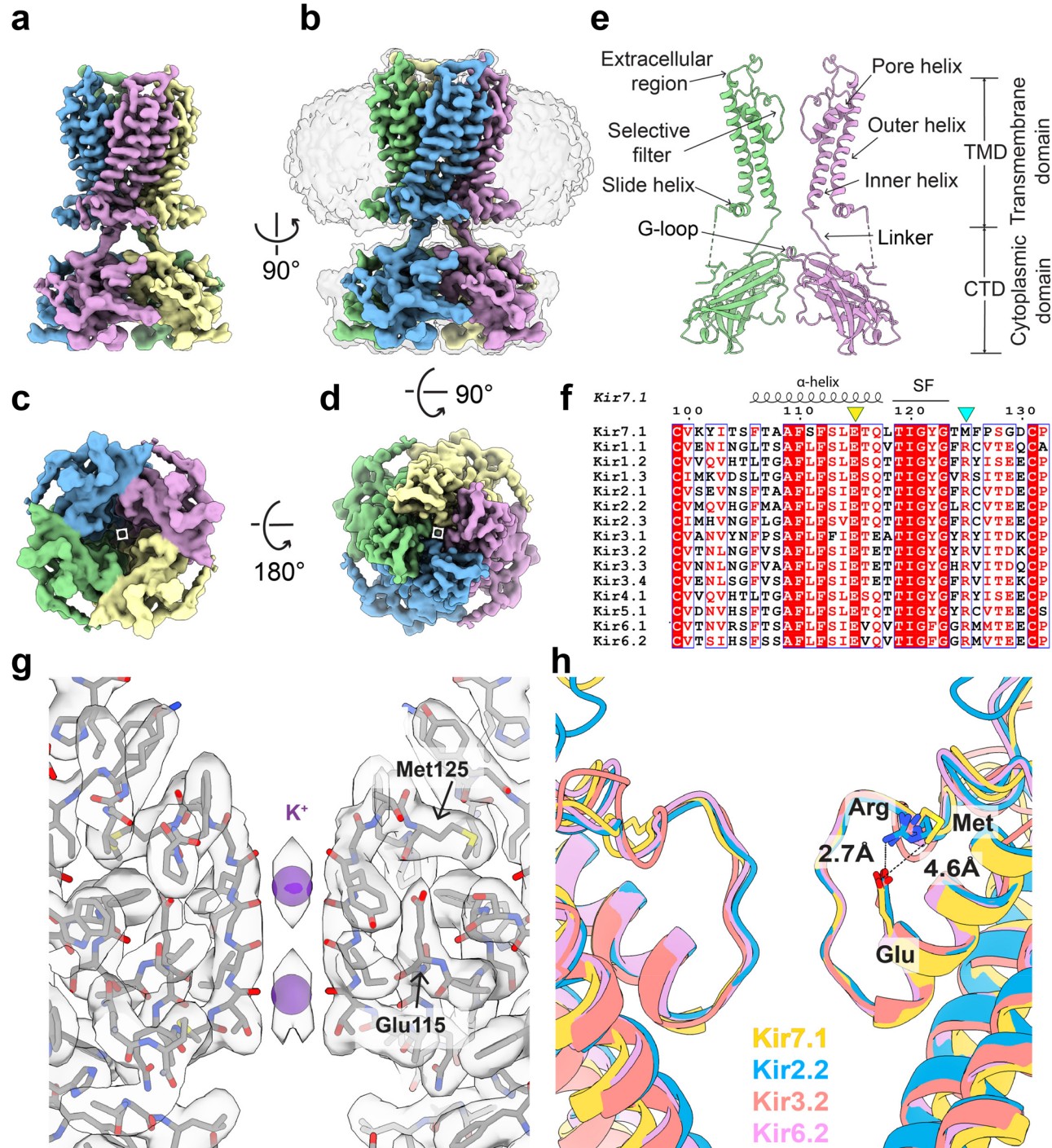

**Fig. 1 | Cryo-EM map and structure of human Kir7.1. a–d** Four views of the cryo-EM map of Kir7.1 obtained at 3.3 Å resolution, shown from the top (**d**) side (**a, b**) and bottom (**c**). The map is scaled to the atomic structure in (**e**) with each subunit color-coded. **e** Side view of the Kir7.1 structure model with two individual monomers colored green and purple (same orientation as (**b**)). Only two chains of Kir7.1 model are shown for clarity. Structural features are labeled, including the outer, inner, pore and side helices, selectivity filter, extracellular region, G-loop, and the linker. Transmembrane (TMD) and cytoplasmic (CTD) domains are indicated. Dashed lines in the model indicate regions where model building was uncertain due to insufficient map resolution. **f** Sequence alignment of the selectivity filter region across all human Kir channels. A Met125 is pointed by a cyan triangle, and its conserved interacting residue Glu115 is labeled in yellow triangle. Selectivity filter is labeled as "SF" at the top. **g** Side view of Kir7.1 selectivity filter fitted into a cryo-EM map with potassium ions (purple spheres) visible inside the pore. Met125 and Glu115 are labeled. **h** Structural alignment of Kir7.1 (yellow) with homologous Kir channels (Kir2.2, blue; Kir3.2, salmon; Kir6.2, pink) highlights conserved architecture in the selectivity filter. A comparison of the Met125 in Kir7.1 versus Arg in other Kir channels is shown on the right. The distance between Glu and Arg in representative Kir2.2 (2.7 Å), is sufficient to form a salt bridge between these residues, however, the replacement of Arg with Met125 in Kir7.1 not only puts this residue 4.6 Å apart, but prevents a salt bridge formation due to a nature of thioether side chain.

**Table 1 | Cryo-EM data collection, refinement, and validation statistics**

| | (1) Extended | (2) Docked | (3) Drug-bound |
|---|---|---|---|
| Data collection and processing | | | |
| Krios microscope location | MIZZOU | CWRU | NCCAT |
| Direct Detector | Falcon4 | K3 | Falcon4 |
| Data collection software | EPU | SerialEM | Leginon |
| Cs (spherical aberration) (mm) | 2.7 | 2.7 | 2.7 |
| Magnification | 130,000 × | 105,000 × | 165,000 × |
| Pixel size (Å) | 0.9525 | 0.828 | 0.7304 |
| Total electron dose ($e^-/Å^2$) | 40 | 50 | 50 |
| No. of images | 12,677 | 5,666 | 12,187 |
| Defocus range (μm) | −0.4 to −2.8 | −0.6 to −2.5 | −0.6 to −2.5 |
| Symmetry imposed | C4 | C4 | C4 |
| Particle images (no.) | 109,323 | 146,544 | 90,037 |
| Map resolution (Å) | 2.8–3.3 | 2.8–3.9 | 2.8–4 |
| FSC threshold | 0.143 | 0.143 | 0.143 |
| Refinement | PDB: 9PR5 EMD-71798 | PDB: 9PR6 EMD-71799 | PDB: 9PR7 EMD-71800 |
| Model composition | | | |
| Non-hydrogen atoms | 9668 | 8936 | 9680 |
| Protein residues | 1180 | 1104 | 1180 |
| Ligands | $PIP_2$, cholesterol | $PIP_2$ | Ent-17OHPC, diC8-$PIP_2$ |
| B factors ($Å^2$) | | | |
| Protein | 94.53 | 68.85 | 111.54 |
| Ligand | 98.18 | 130.87 | 104.25 |
| R.m.s. deviations | | | |
| Bond lengths (Å) | 0.002 | 0.002 | 0.003 |
| Bond angles (°) | 0.453 | 0.397 | 0.654 |
| Validation | | | |
| Correlation coefficient (CCmask) | 0.69 | 0.66 | 0.77 |
| MolProbity score | 1.03 | 1.23 | 1.7 |
| Clashscore | 1.25 | 3.21 | 15.73 |
| Poor rotamers (%) | 0.77 | 0.42 | 0.00 |
| Ramachandran plot | | | |
| Favored (%) | 96.89 | 97.37 | 98.62 |
| Allowed (%) | 3.11 | 2.63 | 1.38 |
| Disallowed (%) | 0.00 | 0.00 | 0.00 |

and inositol ring fit well with in the cryo-EM map, and the binding environment involved conserved amino acids coordinating three phosphate headgroups: His26, Arg54, Lys159, Arg162 and Lys164 (Fig. 3b, c). However, N-terminal His26 coordination of 4-phosphate is distinct for Kir7.1, as this group is usually coordinated by Lys64 in other Kir channels, such as Kir3.2[21] (Fig. 3b, c). This substitution may weaken the interaction with $PIP_2$ and could contribute to the increased flexibility of the CTD, facilitating the large-scale 45° rotation observed between channel states.

## Synergetic effect between steroids and phosphoinositides on Kir7.1 activation

Our previous work demonstrated that progesterone (P4) and DHEA acted as potent endogenous activators of mammalian Kir7.1[16,19], while estrogen, testosterone, and cortisol were inactive[19]. This discovery of endogenous steroid activators of Kir7.1 posed a question whether these steroids cooperate with $PIP_2$ in modulating Kir7.1 activity. To answer this, we performed whole-cell patch clamp recordings under two distinct conditions as described below (Fig. 3d–f). First, we recorded Kir7.1 current in the absence of endogenous $PIP_2$ achieved by its sequestration with membrane permeable $PIP_2$-binding peptide (PBP) that was added extracellularly. This was followed by P4 application (Fig. 3e). Second, we recorded Kir7.1 currents in the presence of synthetic $PIP_2$ (diC$_8$-$PIP_2$) delivered intracellularly via pipette solution, also followed by P4 application (Fig. 3f). It was determined that P4 alone is sufficient to stimulate Kir7.1 activity, even in the absence of $PIP_2$. However, when both $PIP_2$ and P4 were present, the channel exhibits significantly enhanced activation, indicating a synergistic effect between two agonists (Fig. 3d). These findings support a model in which $PIP_2$ binding is not strictly required for channel opening but rather enhances the efficacy of steroid-mediated activation—possibly by stabilizing a conformation more favorable for steroid binding or gating.

## Endogenous $PIP_2$ lipids are present in both docked and extended states

In several Kir channels, i.e., Kir2.2[25,26], the binding of $PIP_2$ alone is sufficient for a significant conformational change of CTD, ensuring conductive state of the channel. However, Kir7.1 may not follow this trend. Interestingly, detailed analysis revealed, that $PIP_2$ density corresponding to the endogenous $PIP_2$ appeared at the same location in both E- and D-states (Figs. 3a, 4a, d, e and S4i, j). However, while two of the $PIP_2$ phosphate groups were coordinated with the same amino acids, i.e., Arg54, Lys159, Arg162, and Lys164, in both conformations, the third $PIP_2$ phosphate headgroup was coordinated by different aminoacids in E- and D-states (Figs. 3a, b and 4a, d, e). Notably, in the D-state this headgroup was associated with N-terminal His26. However, in the E-state, the same phosphate headgroup formed a salt bridge with $NH_2$-moiety provided by Lys195 (Fig. 4e) due to CTD descending away from TMD, and hence moving His26 further away from the interaction site (Fig. 3a, b).

The fact that $PIP_2$ was present in both states despite their clear conformation differences implies that an additional factor may be responsible for a transition between E- and D-states. Indeed, further structural alignment between E- and D-states reveals an additional non-protein density localized between inner and outer helixes of the adjacent Kir7.1 monomers near the extracellular leaflet (Fig. 4b). This density was prominent in the E-state and absent from the D-state, with the shape resembling a steroid-like molecule. Consequent docking analysis against several common steroids, revealed the best fit for the cholesterol (Figs. 4a, b and S4g). The binding pocket surrounding the cholesterol density within a 3.5 Å radius is formed by residues Phe110, Thr107, and Ile140, with Phe106 and Phe110 engaging in $\pi-\pi$ interactions with the steroid rings. Additionally, the OH-moiety of Thr107 is positioned within 5.04 Å from cholesterol OH-group, which may form weak hydrogen interaction only in the aquatic environment (Fig. 4b, c).

To investigate potential influence of cholesterol on channel dynamics, we performed whole-cell patch-clamp recordings using a HEK293 cell line stably expressing Kir7.1. The channel was first activated by progesterone (P4), followed by application of an excess concentration of water-soluble cholesterol (CHL; Figs. 4f and S8a, b). When cholesterol was applied at a tenfold molar excess over progesterone, a clear inhibition of Kir7.1 current was observed (Fig. S8a–c). In contrast, when progesterone was present at a higher concentration than cholesterol, it effectively prevented cholesterol-induced inhibition (Figs. 4g and S8b, c). To assess whether two steroids compete with each other, we measured dose-response of Kir7.1 activation by progesterone in the absence and presence of 30 μM cholesterol (Fig. 4g). Indeed, cholesterol presence significantly shifted progesterone $EC_{50}$

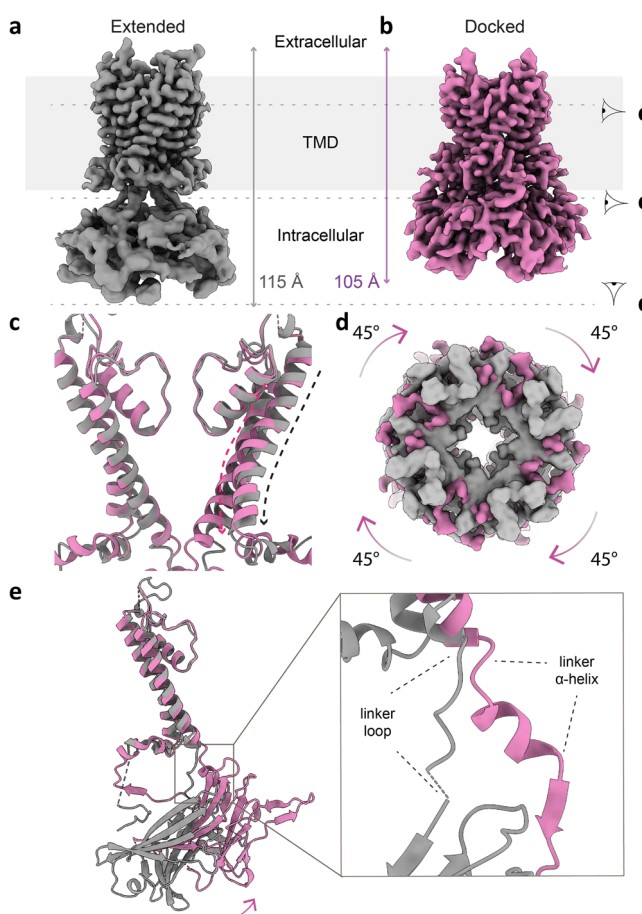

**Fig. 2 | Human Kir7.1 adopts two distinct conformational states. a** Cryo-EM density of Kir7.1 in the extended (E-state) resolved at average resolution of 3.3 Å (grey). The transmembrane region is highlighted by a shaded background. **b** Cryo-EM density of Kir7.1 in the docked (D-state), resolved at average resolution of 3.9 Å (pink), shown at the same scale and view as E-state on the left. Total length of the complex in each state is labeled with its corresponding color. Extended map and docked map are aligned in cryoSPARC[40] and their length were roughly estimated by using scale bar, showing extended map 115 Å and docked map 105 Å. Reference planes (**c**–**e**) correspond to panels below. **c** Structural alignment of the transmembrane domains between the E-state and D-state reveals a conserved selectivity filter but a curvature shift in the outer helix. **d** Superimposed cryo-EM density maps of the CTD in both states, viewed from the intracellular side, showing 45° clockwise rotation of the CTD in the D-state relative to the E-state. **e** Side view of the aligned Kir7.1 models highlighting the conformational rearrangement of the linker between CTD and TMD. The insert shows the structural transition of the linker region from a flexible loop (E-state) to an α-helix (D-state), suggesting its role in mediating domain movement.

from 12 to 26 μM (Fig. 4g). These data indicate that cholesterol exhibits inhibitory effect on Kir7.1 and stabilizes an inactive conformation of the channel, which is further confirmed by the absence of cholesterol-like density in the D-state.

In summary, the additional cholesterol density observed in the E-state (Fig. 4b) and its absence from the D-state supports a model in which full channel activation may require both PIP₂ binding and partial or complete displacement of inhibitory lipids, such as cholesterol, by activating steroids to favor a conductive conformation.

## Activation of Kir7.1 by synthetic steroids and stereoisomers

To better understand the structure-functional relationship between steroid structure and channel activation, we analyzed a panel of synthetic steroids, endogenous steroids, and their stereoisomers (Figs. 5a, b and S9). This screen identified additional activators of Kir7.1,

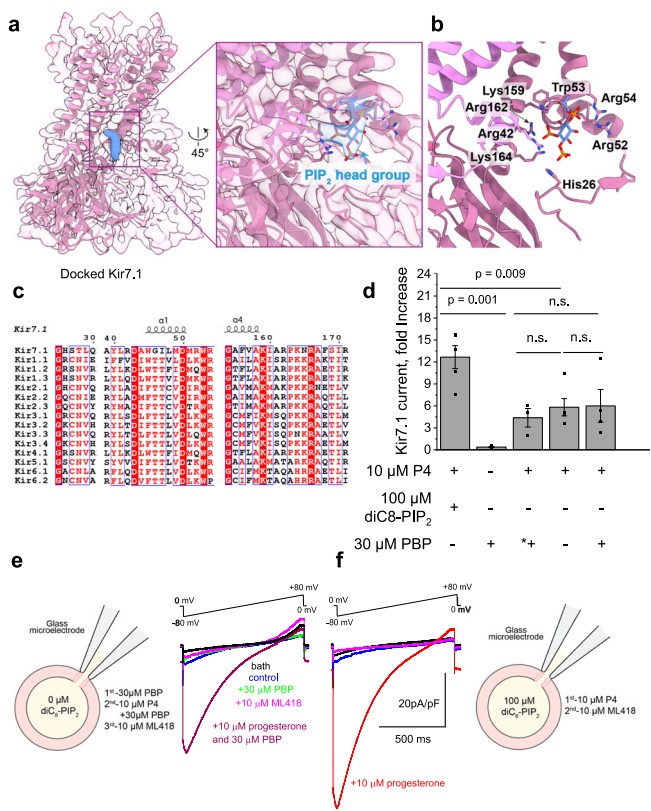

**Fig. 3 | Endogenous PIP₂ is associated with human Kir7.1 in the docked (D-state) conformation and synergizes with progesterone to enhance channel activity. a** Side view of the cryo-EM density map of Kir7.1 in the D-state (purple), showing a prominent non-protein density corresponding to bound PIP₂ (blue). Inset: zoomed-in PIP₂ density. Maps and models are rotated by 45° for clarity. Endogenous PIP₂ is partially modeled where it fits the density. **b** Close-up of the PIP₂ binding pocket, highlighting positively charged residues within 3.5 Å of PIP₂ (Lys159, Arg162, Arg42, Lys164, Arg52, Arg54, and His26). PIP₂ is shown in blue and colored by atom (oxygen, red; nitrogen, dark blue; phosphorus, orange). **c** Sequence alignment of conserved PIP₂-binding motifs across human Kir channels, highlighting conserved residues coordinating phosphoinositide binding. Kir7.1 is shown at the top. Secondary structure elements (α-helices) are annotated, and conserved residues are boxed in red. Alignment was generated using Jalview and rendered with ESPript. **d** Quantification of Kir7.1 fold activation by progesterone (P4) in the presence or absence of PIP₂. Currents at −80 mV were normalized to baseline (control, no P4) recordings. No statistical differences (n.s.) were noted between conditions when progesterone was applied alone (P4+, PBP−), with PBP (P4+, PBP+), or consequently, immediately after PBP administration (P4+, PBP*+). Data are mean ± S.E.M., n = 3–7 cells. Post hoc Tukey's tests were used for comparisons of mean values and statistical significance. **e** Representative whole-cell recordings from HEK293T cells expressing Kir7.1. First, application of the PIP₂ binding peptide (PBP, green) to sequester endogenous PIP₂ pool notably inhibited Kir7.1 responses. No synthetic diC8-PIP₂ was applied. Second, Kir7.1 was further stimulated with 10 μM progesterone (P4, plum) which led to a strongly increased Kir7.1 current, indicating that progesterone can activate the channel without PIP₂. Bath (black) indicates recordings in standard Krebs solution; control (blue) denotes recordings in KMeSO₃-based external solution. **f** Kir7.1 was first stimulated with 10 μM progesterone (P4, red), concurrently with 100 μM of synthetic intracellular diC8-PIP₂, producing an even stronger Kir7.1 activation, indicating a cooperativity between PIP₂ and progesterone. In both cases, ML418 (magenta), Kir7.1 inhibitor, was applied at the end of the recordings to return to the baseline.

including enantiomer of progesterone (ENT-progesterone or ENT-P4), enantiomer of 17-hydroxyprogesterone caproate (ENT-17OHPC), pregnenolone, and a progestin 17α-methylprogesterone (MQ351; Figs. 5a–c and S9). Subsequent dose–response analyses revealed that these compounds act within the low micromolar to submicromolar concentration range and exhibit altered on/off kinetics, suggesting

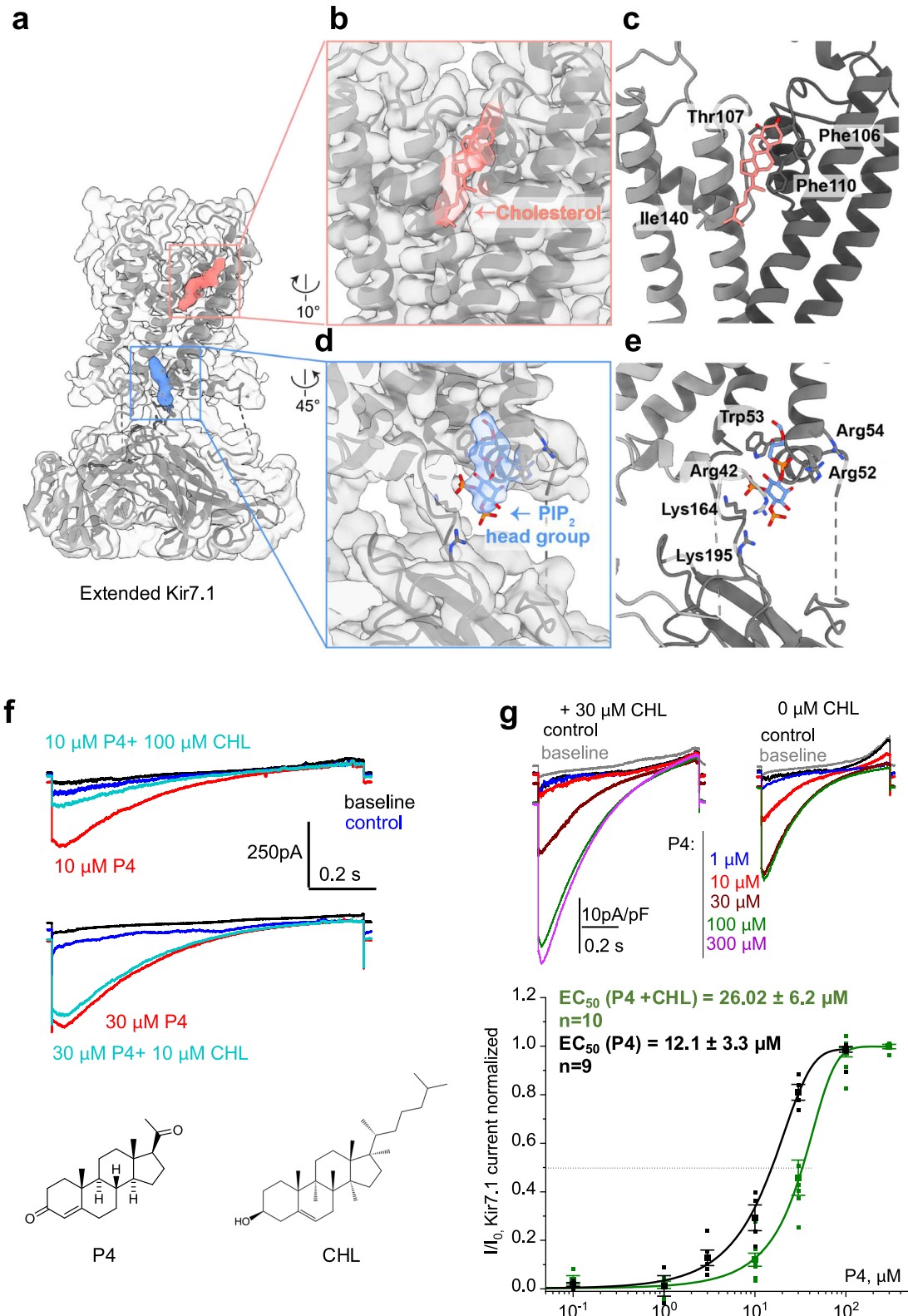

sustained modulation of channel activity (Figs. 5c, d and S10). Specifically, ENT-17OHPC emerged as the most potent activator of Kir7.1, demonstrating EC₅₀ of 690 nM (Fig. 5d) and inability of the activated channel to be blocked by ML418 (Figs. 5c and S10), a Kir7.1 antagonist[27]. Interestingly, the presence of a caproate ester group on a 17α-hydroxyl group as in 17OHPC and its enantiomer resulted in stronger affinity and delayed off-rates. Notably, steroids containing a double bond at the

position of carbon 5 linking the steroid A- and B-rings showed a variety of activation potencies with pregnenolone emerging among the most potent (Fig. 5b, e), in addition to ENT-17OHPC and ENT-P4. However, the reduction of this double bond produced steroids with a fully saturated and rigid A- and B-rings with almost no activity toward Kir7.1, as demonstrated by allopregnanolone and its analogues (Figs. 5b, e and S9).

**Fig. 4 | PIP₂ and cholesterol bind to human Kir7.1 channel in the extended (E)-state. a** Side view of the cryo-EM map of the Kir7.1 channel in the extended (E-state) conformation (light gray), with a PIP₂ molecule bound at the conserved interaction site (blue), and a distinct cholesterol (salmon) sits between two monomers. **b** Zoomed-in side view of E-state Kir7.1 (light gray) showing an additional non-protein density modeled as cholesterol (salmon). Maps and models are rotated by 10° for clearer visualization in (**b**) and (**c**). **c** Close-up view of the cholesterol-binding site between helices of adjacent subunits (left monomer-light gray; right monomer-dark gray). Cholesterol is shown with oxygen atoms in red and carbon in salmon. **d** Close-up view of the PIP₂ molecule (blue) bound in E-state (gray), colored by atom type: oxygen (red), nitrogen (blue), and phosphorus (orange). 45° rotation is performed for clearer visualization in (**d**) and (**e**). The structure of endogenous PIP₂ is partially modeled in the region to where it fits the density. **e** In PIP₂ binding pocket, positively charged residues (Arg42, Arg52, Arg54, Lys164, and Lys195) involved in PIP₂ coordination are highlighted. **f** Representative whole-cell recordings showing the effect of cholesterol (CHL) on P4-induced Kir7.1 activation. Top: 10 μM P4-induced activation is reduced by co-application of 100 μM CHL. Bottom: 30 μM P4 largely overcomes the inhibitory effect of 10 μM CHL. Chemical structures of progesterone (left) and cholesterol (right) are shown below. **g** Upper panel: representative Kir7.1 potassium currents ($I_{K+}$) recorded from hKir7.1-expressing stable cell line (HEK293T-Kir7.1) in response to either P4 or P4 + CHL. Lower panel: Dose-response of P4 in the absence (black, $n = 10$) and the presence of 30 μM CHL (green, $n = 9$). $EC_{50}$ values were calculated from nonlinear regression fits. Data are presented as mean ± S.E.M. ($n$ = number of cells used per condition).

## Structural analysis of the steroid-binding pocket and permeability pathway

To better understand the association between Kir7.1 and steroids, we purified the ENT-17OHPC-bound form of Kir7.1 using a procedure similar to the employed one for the D- and E-state purifications, with the key difference that ENT-17OHPC and synthetic diC8-PIP₂ were added to the purified protein immediately prior to vitrification (Fig. 6a). ENT-17OHPC -bound structure of the human Kir7.1 channel was resolved by single-particle cryo-EM to an overall resolution of 4.0 Å for C1 symmetry and 3.4 Å for C4 symmetry (Figs. 6a, b, S4c, f, h, k, S11 and Table 1). As anticipated, the density corresponding to ENT-17OHPC was resolved in the same hydrophobic pocket as the distinctive non-protein density attributed to cholesterol—positioned between the inner and outer helices near the extracellular side of the membrane (Fig. 6b, c) surrounding by residues within a 3.5 Å radius including: Thr107, Phe106, Phe110, Val65, Ser61, and Ile140 (Fig. 6c). These residues form a hydrophobic pocket, consistent with the steroid scaffold, while Thr107 provides polar moiety to coordinate ENT-17OHPC's ketone group.

The synthetic phosphoinositide diC8-PIP₂ was also resolved and anchored by a cluster of similar conservative positive residues (Arg52, Arg54, Arg42, Lys159, Lys164, and Trp53), as previously observed in E-state (Fig. 6d, e). These interactions resembled typical Kir-family PIP₂ coordination and showed that ENT-17OHPC does not displace PIP₂ from its binding site but rather occupies the same steroid binding cavity previously occupied by a cholesterol (Fig. 6b, c).

Structural superposition of the ENT-17OHPC-bound state and E-state showed that this compound positions its ketone group closer to Thr107 OH-moiety. As mentioned above, Thr107's hydroxyl group was located within 5.04 Å from OH-group of cholesterol, a relatively long distance to form strong hydrogen bond interaction. However, in the ENT-17OHPC-bound state, Thr107 OH-moiety is located only 2.76 Å away from its ketone group ensuring sufficient hydrogen bond formation (Fig. 6f–h). Which in its turn will stabilize association between ENT-17OHPC and the channel, and anchor the compound at the interface of hydrophilic and hydrophobic cavities.

A comparison between ENT-17OHPC and cholesterol structures indicate their distinct polarity profiles, i.e., ENT-17OHPC presenting two hydrophilic zones, while cholesterol presenting only a single one (Fig. 6h). These properties would require ENT-17OHPC to engage both hydrophobic and polar contacts, whereas cholesterol interacts almost exclusively rely on nonpolar contacts. In the ENT-17OHPC-bound conformation, the same binding pocket allows both hydrophilic interaction between compound's ketone and Thr107, and nonpolar interactions between ENT-17OHPC lipophilic tail and alpha-helix of the M2 region. Such binding would allow ENT-17OHPC to displace cholesterol from the binding site.

Further comparison of the ion conduction pathways between the E-, D-, and ENT-17OHPC-bound states using the HOLE2 program[28] based on solvent-accessible (water-filled) cavities revealed key gating differences (Fig. 7a–d). In both E- and ENT-17OHPC-bound states, the channel exhibits a relative constricted inner helix gate (6 Å; Figs. 7a, c and 8) with a relaxed G-loop gate (9.4 Å to 10.15 Å; Figs. 7a, c and 8), and an elongated ion passage, reducing the efficiency of ion permeation. Upon transition to the D-state, the conformation shifts, resulting in a relaxation of the inner helix gate (10 Å; Figs. 7b and 8). However, this "docked" conformation is still unlikely to result in the "open" state to ensure ion permeation due to a constriction of the passage formed at the G-loop gate by Ile281 (8.35 Å; Figs. 7b and 8). In the D-state, electrostatic attraction from the nearby Glu280 residues can entrap a hydrated $Mg^{2+}$ ion (8.6 Å[29] in diameter; Fig. 8). In fact, $Mg^{2+}$-block and consequent inward rectification is a prominent feature of Kir-channel family[30]. Interestingly, in the ENT-17OHPC–bound state, the passage expands to 10.15 Å in diameter (Fig. 8), a change that likely relieves the $Mg^{2+}$ block and facilitates potassium efflux toward the extracellular side during membrane depolarization.

All three conformational states adopt certain constrains preventing a full open conformation, i.e., E- and ENT-17OHPC-bound states showing constricted inner helix gate, while the permeation in D-conformation is likely impeded by Mg-block. Therefore, we hypothesize that the bona fide open conformation is likely an intermediate state between D- and ENT-17OHPC with relaxed, ~ 11 Å G-loop, and an open inner helix gate ~ 10 Å (Fig. 8).

In conclusion, our structural analysis of human Kir7.1 uncovers key mechanistic insights into its regulation by endogenous bioactive lipids, including $PI_{4,5}P_2$ and steroid modulators. Beyond delineating the gating transitions of the channel, our findings reveal a distinct allosteric mechanism that enables cooperative channel activation. We propose a model in which endogenous cholesterol functions as an inhibitory ligand that can be competitively displaced by select activating steroids. These steroids, acting in synergy with $PI_{4,5}P_2$, facilitate the full opening of the channel. Additionally, we report the identification of several small-molecule activators of Kir7.1, offering powerful tools to investigate and therapeutically target this essential ion channel in both physiological and disease states.

## Discussion
This study presents several high-resolution structures of human Kir7.1 captured in multiple conformational states, providing critical insight into the molecular mechanisms underlying its gating and regulation. We resolved Kir7.1 in three distinct structural conformations that likely correspond to key physiological states of the channel. The common structural features among all three conformations are: similar width of selectivity filter, and relatively wide inner helix gates > 3 Å radii. The area undergoing conformation changes are restricted to: inner helix gates, linker regions and CTDs.

The extended conformation (E-state) represents a putative non-conducting state of the channel and is characterized by the binding of cholesterol and endogenous phosphatidylinositol 4,5-bisphosphate (PIP₂). Two structural features likely interfere with ion conductivity in this state. First, a hydrophobic region surrounding the G-loop, constrained by Ile281 near the cytosolic entrance of the pore to 9.44 Å, may impede cation entry. Second, the extended conformation elongates the ion conduction pathway, stretching the channel cavity to approximately 115 Å, which could significantly slow ion permeation (Fig. 8, gray model).

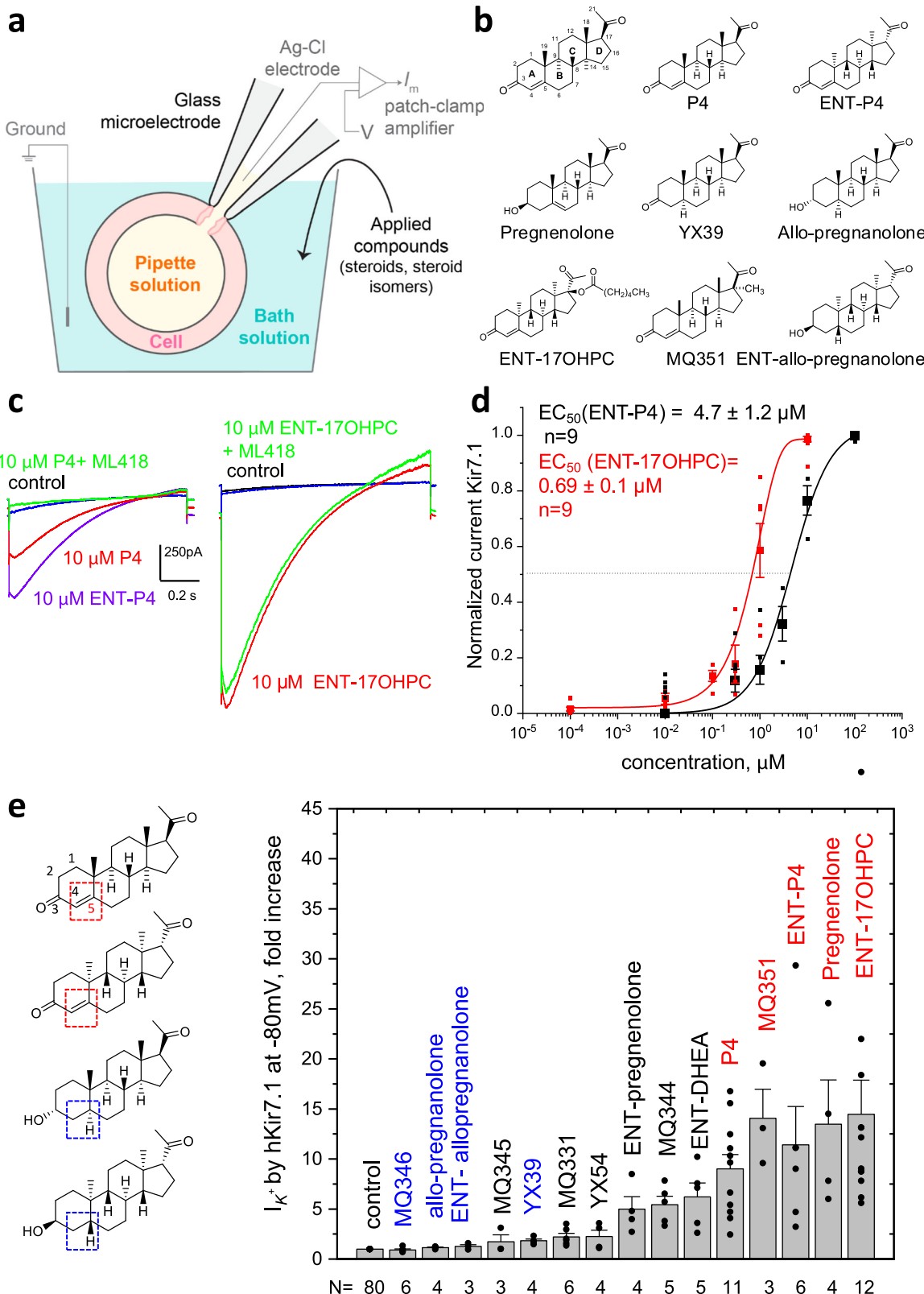

Detection of cholesterol in the E-state revealed a hydrophobic pocket, likely serving as a steroid-binding site on Kir7.1. Notably, removal of cholesterol−while retaining $PIP_2$ binding−induced a significant conformational change in the channel, resulting in the docked conformation (D-state). This state represents a semi-open configuration stabilized by interactions with $PIP_2$ (Fig. 8 pink model). However, it features a constricted region at Ile281,

measuring -8.35 Å, which is narrow enough to trap $Mg^{2+}$ ions and cause a $Mg^{2+}$-dependent block−underlying the inward rectification typical of the Kir channel family. Under physiological conditions, this conformation would hinder outward potassium flow in the D-state.

Another notable conformational change during the transition from the E- to D-state occurs at the linker region between the CTD and

**Fig. 5 | Kir7.1 is activated by select synthetic steroids. a** Schematic illustration of the whole-cell patch-clamp configuration used for testing the indicated compounds under electrophysiological recording conditions. **b** Chemical structures of endogenous and synthetic steroids along with their stereoisomers, tested for Kir7.1 activation by whole-cell patch clamps in HEK293T cells. A progesterone molecule with carbon atom numbered, and steroid rings labeled is shown. **c** Dose-response curves of Kir7.1 activation by ENT-P4 and ENT-17OHPC. Left: 10 μM P4 is applied first, followed by 10 μM ENT-P4; Right: 10 μM ENT-17OHPC is applied. **d** Representative $I_{K+}$ mediated by human Kir7.1 expressed in HEK293T cells in response to progesterone (P4), ENT-progesterone (ENT-P4), and ENT-17OHPC.

ML418 was applied at the end of each recording to confirm current identity. $EC_{50}$ values were calculated from nonlinear regression fits. Data are presented as mean ± S.E.M. ($n = 9$ cells per condition). **e** Fold increase in $I_{K+}$ mediated by human Kir7.1 in response to the indicated compounds. $N$ indicates the number of cells recorded per condition. Data are presented as mean ± S.E.M. Red boxes highlight the important double bond in steroid backbone essential for compound activity. Strong agonists are labeled in red. Blue boxes highlight a saturated A-ring, and the absent double bond in theinactive compounds labeled in blue. Compounds in black showed intermediate activity.

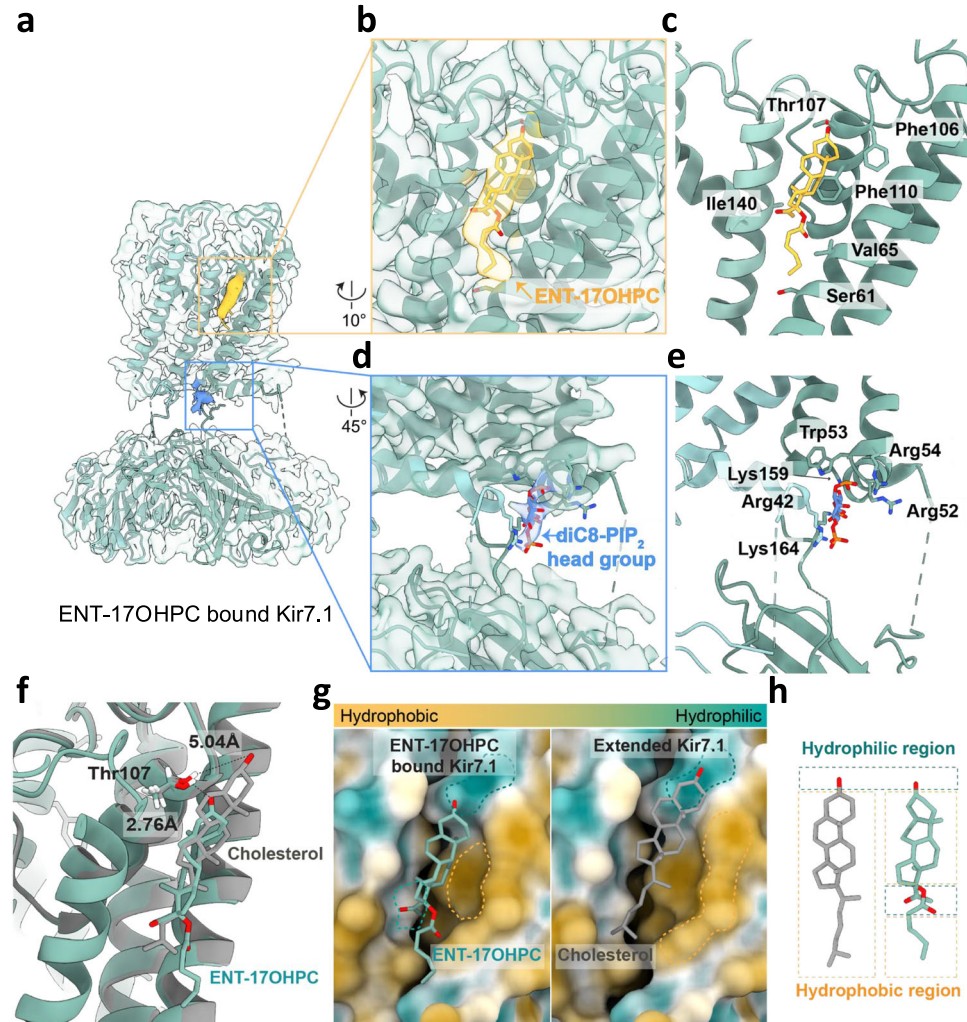

**Fig. 6 | Enantiomer of 17-OHPC (ENT-17OHPC) bound state Kir7.1 structure. a** Side view of the cryo-EM map of the Kir7.1 channel in the ENT-17OHPC-bound state (teal), with a diC8-PIP₂ molecule bound at the conserved interaction site (blue), and ENT-17OHPC (yellow) positioned between two monomers. **b** Zoomed-in side view of ENT−17OHPC-bound Kir7.1 (teal) showing an additional non-protein density modeled as ENT−17OHPC (yellow). Maps and models in (**b**) and (**c**) were rotated by 10° to improve visualization. **c** Close-up view of the ENT−17OHPC binding site between helices of adjacent subunits (left monomer, light teal; right monomer, dark teal). ENT-17OHPC is shown with oxygen atoms in red and carbon atoms in yellow. **d** Close-up view of the diC8-PIP₂ molecule (blue) bound in ENT-17OHPC-state (teal), colored by atom: oxygen (red), nitrogen (blue), and phosphorus (orange). Maps and models are rotated 45° for clearer viewing in (**d**) and (**e**). The diC8-PIP₂ model is partially modeled in the region to where it fits the density. **e** In diC8-PIP₂ binding pocket, positively charged residues (Arg42, Arg52, Arg54,

Lys159, and Lys164) involved in PIP₂ coordination are highlighted. **f** Comparison of steroid binding sites between the extended Kir7.1 conformation and the ENT-17OHPC-bound Kir7.1 structure. Cholesterol is shown with oxygen atoms in red and carbon atoms in gray. ENT-17OHPC is colored with oxygen atoms in red and carbon atoms in cyan. **g** Surface hydrophobicity of the Kir7.1 cavity is shown, colored from hydrophobic (orange) to hydrophilic (teal). Left: in the ENT-17OHPC-bound Kir7.1 conformation, ENT-17OHPC (teal) occupies a predominantly hydrophobic pocket adjacent to the pore helices. Right: in the extended Kir7.1 conformation, cholesterol (gray) occupies a similar site that contacts both hydrophobic and polar residues. Dotted outlines indicate the approximate boundaries of each ligand-binding cavity. **h** Schematic of cholesterol (left, gray) and ENT-17OHPC (right, teal) illustrating their hydrophilic (teal box) and hydrophobic (orange box) regions. The oxygen-containing functional groups define the polar hydrophilic regions, whereas the hydrocarbon backbones represent the predominantly hydrophobic regions.

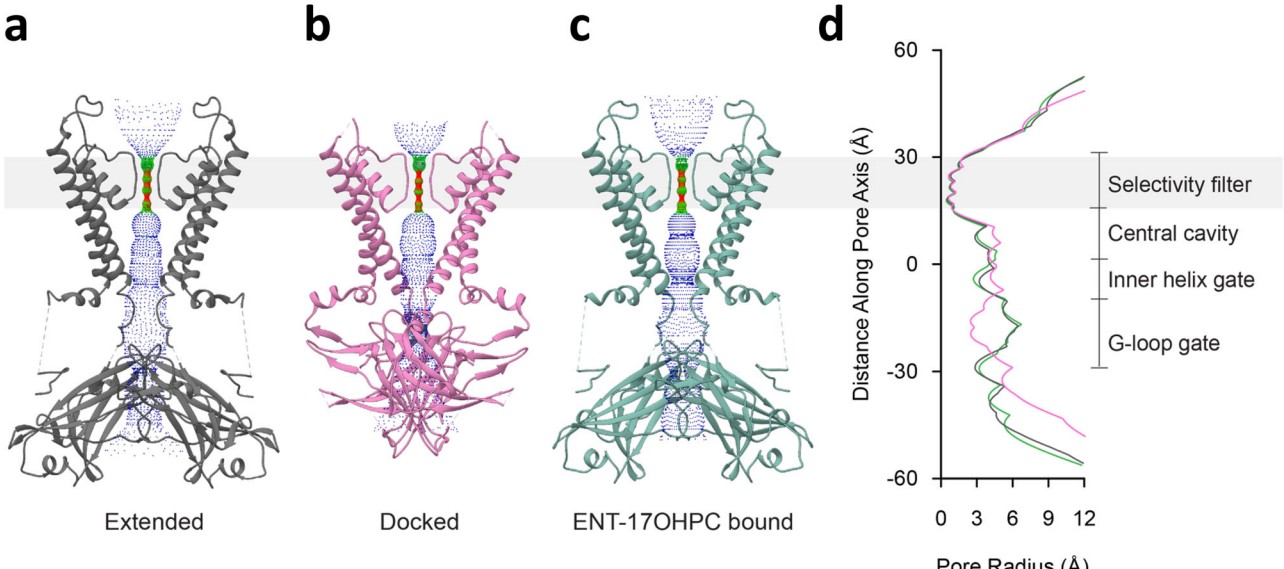

**Fig. 7 | Pore radius analysis and channel conductivity of the human Kir7.1 channel. a–c** Comparative pore radius analysis of the **a** extended conformation, **b** docked conformation, and **c** ENT-17OHPC-bound conformation of Kir7.1, shown with their respective central pore profiles calculated by HOLE2 program. The pore is color-coded according to radius thresholds relative to water molecule size: red indicates a radius too narrow for water permeation, green represents a radius sufficient for a single water molecule, and blue corresponds to regions wide enough to accommodate multiple water molecules. The approximate membrane boundaries are shaded in gray. **d** Plot of pore radii along the ion conduction pathway for each conformation, indicating key structural changes in selectivity filter, central cavity, inner helix gate, and G-loop gate.

TMD. This region undergoes a dynamic shift from a flexible loop in the E-state to a compact α-helix in the D-state (Fig. 2e). Such a transition was predicted in silico for other Kir channels, such as Kir2.1 nearly a decade ago[31], but no experimental evidence has been available until now. Our work provides the long-sought in vitro demonstration of the critical role of this linker region in regulating Kir channel dynamics, supporting the model that CTD movement is essential for transitioning from inactive to active channel states.

Given that the primary difference between the E- and D-states involved the presence of cholesterol—a ligand shown by electrophysiological experiments to inhibit Kir7.1 activity—we proposed a model in which displacement of cholesterol by a homologous steroid could modulate channel gating. Among a panel of tested synthetic steroids, stereoisomer of 17-hydroxyprogesterone caproate, or ENT-17OHPC, emerged as the most potent activator of Kir7.1. The cryo-EM structure of Kir7.1 in the presence of ENT-17OHPC revealed its binding within the same hydrophobic pocket previously occupied by cholesterol, defining a third conformational state: the ENT-17OHPC-bound state (Fig. 8, teal model). Interestingly, this state closely resembled the E-state, with key differences including a slightly widened G-loop gate, reduced hydrophobicity due to Ile281 being positioned ~10 Å apart, and a modestly shortened ion conduction pathway, reducing the channel cavity length to approximately 114 Å (Fig. 8).

Therefore, considering the limitations observed in the resolved structures, we hypothesize that the fully open conformation of the channel should be an intermediate state between the D- and E-states. This putative open state would feature a reduced ion conduction pathway similar to the 105 Å cavity observed in the D-state, a widened G-loop gate, and a lack of magnesium block, thus, collectively supporting efficient ion permeation (Fig. 8, blue model).

The reported structures provide additional insight into the channel ion selectivity. Compared to other members of the Kir family, Kir7.1 exhibits a key structural divergence at position 125, where a methionine (Met125) replaces the highly conserved arginine typically found in this region. This noncanonical substitution likely imparts increased flexibility to the channel's selectivity filter, potentially explaining Kir7.1's ability to conduct cesium ions[1] (Cs+)—a property rare among Kir and other potassium channels, in which cesium commonly acts as a pore blocker.

Moreover, the resolved Kir7.1 structures provide additional insights into the molecular basis of Kir7.1 weak rectification. Among the entire family of Kir ion channels, Kir7.1 is the weakest rectifier, while Kir2.1 is the strongest[32,33], and Kir3.2[21] shows intermediate rectification. This phenomenon can be explained by differences at three structural checkpoints: (1) the CTD entrance; (2) the G-loop gate, and (3) the inner helix cavity (Fig. 9).

At the level of first checkpoint in Kir2.1, i.e., CTD entrance, Asp255 constricts the passage to ~5.6 Å, entrapping hydrated Mg2+ (~8 Å) and preventing K+ permeability (Fig. 9, purple). If partially hydrated Mg2+ (~4.8 Å) passes this gate, it will be further entrapped at the next checkpoint, i.e., the G-loop gate, which is formed by Glu224 and Glu299. These glutamates introduce additional negative charges by lining up 9.7 Å pore and therefore stabilizing Mg2+ within. The final checkpoint, i.e., inner helix, contains Asp172 that provides additional electrostatic anchor for Mg2+. Together, these acidic anchors and tight geometry explain why Kir2.1 nearly abolishes outward K+ flux at depolarized voltages[33].

Similar situation exists in Kir3.2 (GIRK2), with few exceptions[21]. At the CTD entrance, instead of electrostatic trap, Tyr266 and Tyr267 form ~7.5 Å hydrophobic constriction (Fig. 9, orange). The second checkpoint is formed by Phe192, a hydrophobic residue that does not stabilize cations. Finally, at G-loop, the narrowest site is defined by Met313, forming a much wider ~14.6 Å opening, which cannot trap hydrated Mg2+. However, acidic residues in the cytoplasmic vestibule (e.g., Glu236) still provide some stabilization. As a result, Kir3.2 supports inward rectification, but the block is weaker and more transient than in Kir2.1.

However, most acidic anchors outlined above are absent in Kir7.1 (Fig. 9, grey). At the CTD entrance, Asp232 forms a relatively wide ~12.5 Å passage, too wide to effectively coordinate Mg2+. The second checkpoint, the G-loop, is similar in size to Kir2.1 (~9.4 Å), however it formed by nonpolar Ile281, thus, lacking any charge stabilization. The

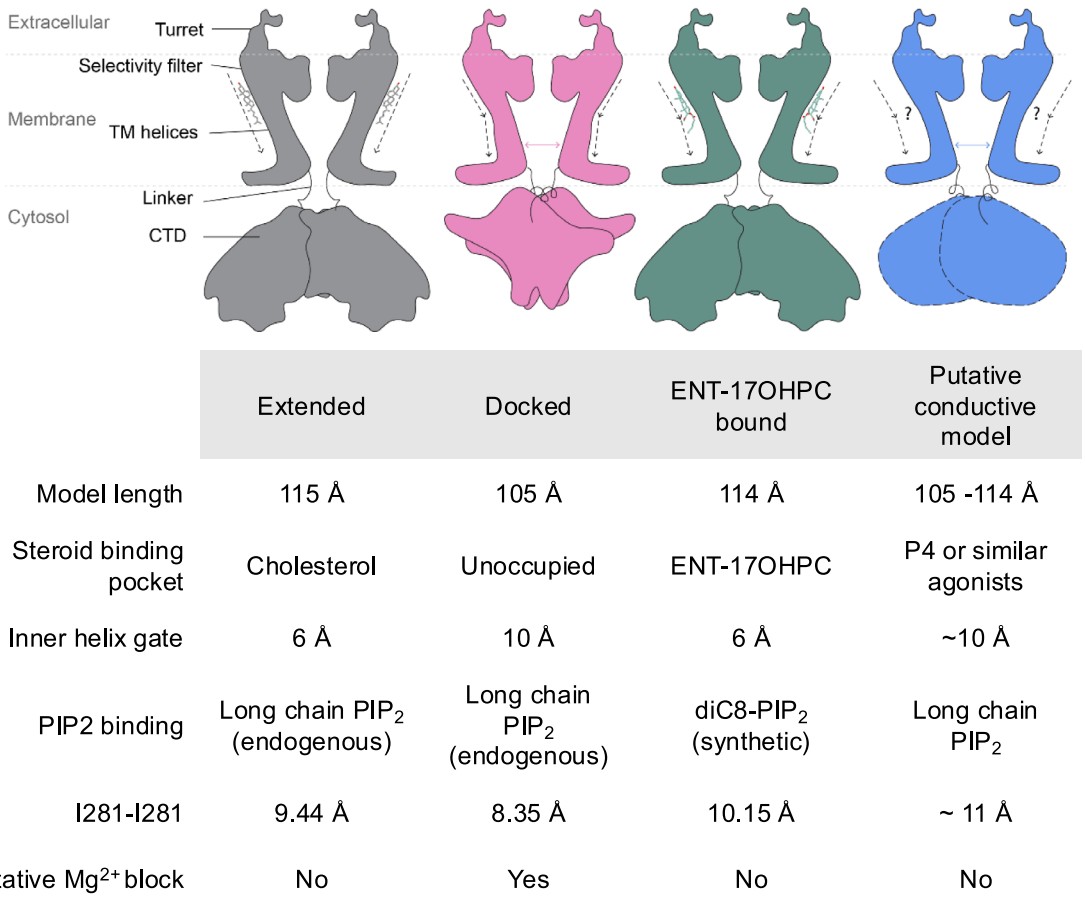

| | Extended | Docked | ENT-17OHPC bound | Putative conductive model |
|---|---|---|---|---|
| Model length | 115 Å | 105 Å | 114 Å | 105 -114 Å |
| Steroid binding pocket | Cholesterol | Unoccupied | ENT-17OHPC | P4 or similar agonists |
| Inner helix gate | 6 Å | 10 Å | 6 Å | ~10 Å |
| PIP2 binding | Long chain $PIP_2$ (endogenous) | Long chain $PIP_2$ (endogenous) | diC8-$PIP_2$ (synthetic) | Long chain $PIP_2$ |
| I281-I281 | 9.44 Å | 8.35 Å | 10.15 Å | ~ 11 Å |
| Putative $Mg^{2+}$ block | No | Yes | No | No |

**Fig. 8 | Schematic illustration of the three resolved cryo-EM conformations of Kir7.1 and the predicted conductive model.** Extended state (gray), docked state (pink), ENT-17OHPC-bound state (teal), and a putative conductive state (blue) are shown to explain the mechanism of the Kir7.1 channel gating. Dashed lines indicate the curvature of the transmembrane helices across different channel states. The question marks in the putative conductive model denote the presence of a hypothetical high-potency agonist that may stabilize this conformation.

final checkpoint at the inner helix gate is represented by another nonpolar residue, Val157. Thus, Kir7.1 allows outward $K^+$ currents with little suppression, producing weakest rectification. In summary, the progressive loss of acidic residues in addition to widening at key checkpoints explains the diversity of rectifications.

A common feature across three resolved Kir7.1 structures is the presence of $PIP_2$ density within the conserved $PIP_2$-binding pocket, observed in both the extended and docked conformations. Interestingly, in most previously reported atomic structures of other Kir channels, $PIP_2$ binding has been associated exclusively with the docked state, and not observed in the extended conformation[20,21,25,34–36]. In fact, the prevailing hypothesis suggests that $PIP_2$ binding alone triggers a conformational rearrangement that draws the cytoplasmic domain upward toward the membrane[20,21,25,34–38]. However, while Kir7.1 docked structure presents a significant 45° clockwise rotation of the CTD, the most dramatic conformational rearrangement among Kir channel family[21], it is puzzling to find $PIP_2$ density in the extended state as well. Three plausible explanations are possible. First, a specific occupancy stoichiometry for $PIP_2$ might be important. It is possible that both E- and D-states can bind $PIP_2$, but only full occupancy of all four channel forming subunits with four $PIP_2$ molecules provides sufficient binding energy to drive the CTD docking transition. Partial occupancy may stabilize the extended conformation while still allowing $PIP_2$ association. Second, an association with diverse phosphoinositide species may occur. Mammalian cells naturally harbor a variety of phosphoinositides, including monophosphorylated species such as $PI_3P$, $PI_4P$, and $PI_5P$, as well as variety of bisphosphates. The structural state of Kir7.1 may be influenced by the specific phosphoinositide bound, and this ligand heterogeneity could account for the coexistence of both extended and docked conformations. Finally, a third and most plausible explanation involves a cooperative interaction between $PIP_2$ and another ligand-binding site, such as steroids. Inhibitory steroids like cholesterol may act as natural ligands or cofactors that stabilize the CTD in the extended state, thereby maintaining a closed channel conformation even in the presence of bound $PIP_2$. We propose that cholesterol binding increases the rigidity of the transmembrane helices, preventing them from turning inward and thereby restricting the movement required for CTD lifting and compaction (Fig. S7b). In contrast, when $PIP_2$ is present and the steroid binding pocket (SBP) is unoccupied, the TM helices can move more freely. Only when the helices compress to an appropriate degree is sufficient space created for CTD rotation and docking. Thus, cholesterol binding effectively locks the helices at a fixed angle, straightening the outer helix and stabilizing the extended conformation by preventing the "kinked" flexibility that normally facilitates CTD docking. Consistent with this interpretation, structural analysis shows that in the absence of bound steroids, the SBP remains flexible, allowing the outer helix to bend at a larger angle.

Therefore, cholesterol removal could be essential for enabling the conformational transition between the extended and docked states. However, in the content of a living cell, complete cholesterol sequestration is impossible, unless it is replaced by other steroid-like entity. Hence, we propose that a replacement of cholesterol with activating

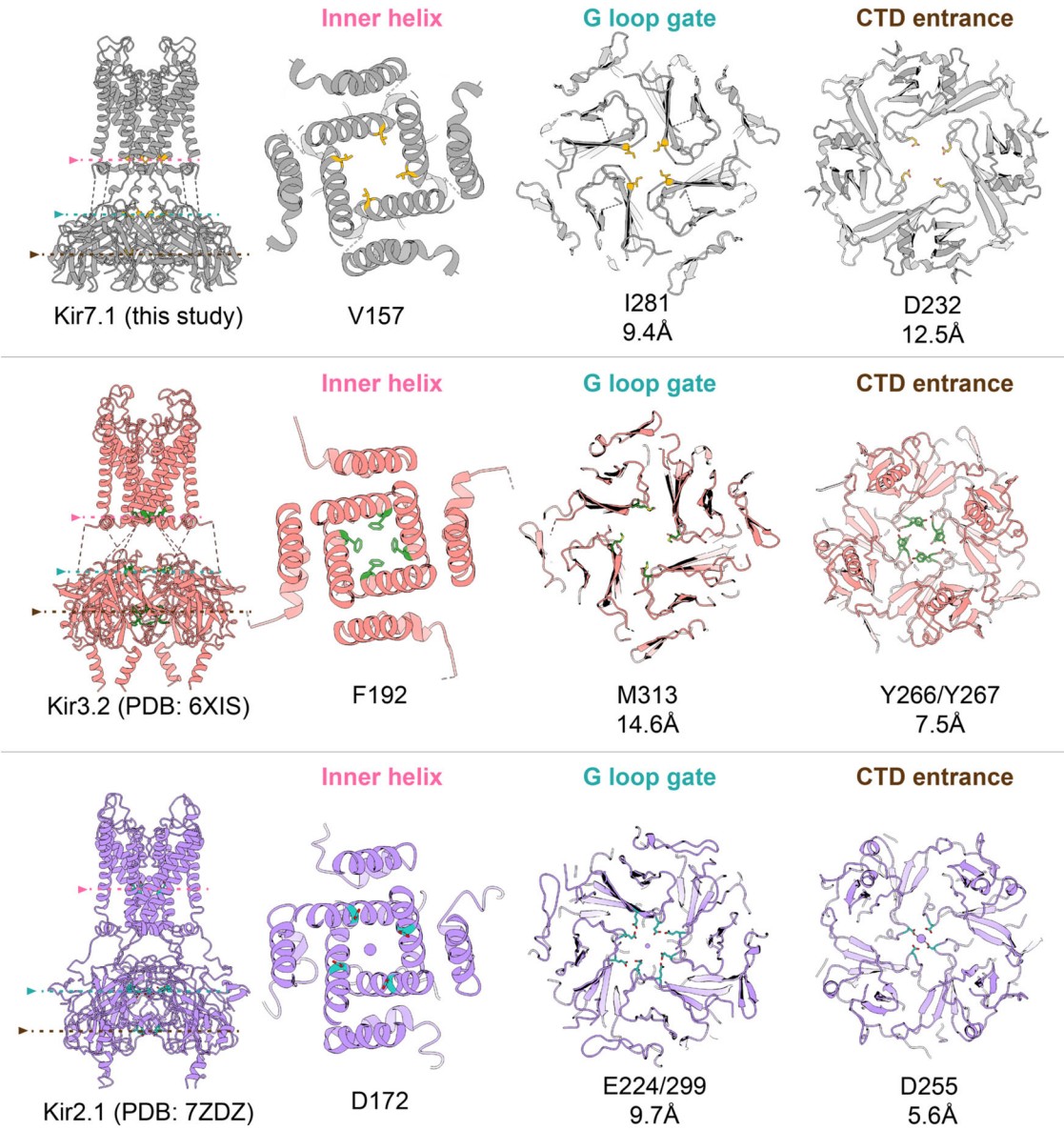

**Fig. 9 | Structural comparison of Kir7.1 (gray), Kir3.2 (orange), and Kir2.1 (purple) highlighting key determinants of inward rectification strength.** Cryo-EM structures of representative Kir channels are shown with detailed views of three structural regions that contribute to inwardly rectification: the inner helix (pink), the G-loop gate (teal), and the CTD entrance (brown). Top: Kir7.1 (this study, PDB: 9PR5, gray). Middle: Kir3.2 (PDB: 6XIS; orange). Bottom: Kir2.1 (PDB: 7ZDZ; purple). V Valine (Val), I Isoleucine (Ile), D Aspartate (Asp), F Phenylalanine (Phe), M Methionine (Met), Y Tyrosine (Tyr), E Glutamate (Glu).

steroids, such as progesterone, pregnenolone, or ENT-17OHPC, facilitates a transition to an open state. By occupying SBP, these activating steroids would reduce the rigidity of the transmembrane helices observed in the E-state, and allow intermediate CTD rotation (Fig. 8). Such intermediate rotation should open both gates and produce a conductive state of the channel as illustrated on Fig. 8 (blue model). We suspect that a conformation, currently represented by ENT-17OHPC-bound state, captures a channel in a primed but not yet conductive state. This primed conformation could result from the presence of synthetic $PIP_2$ used during purification. By competing with endogenous long-chain $PIP_2$, synthetic short-chain $PIP_2$ provides weaker anchoring strength for CTD docking. Another possibility is a presence of glyco-diosgenin (GDN) detergent, a steroid-derived amphiphile, that may compete with ENT-17OHPC for the same binding pocket. In such scenario, only a portion of four SBP in the homo-tetramer are occupied by ENT-17OHPC, while other are occupied by GDN, therefore preventing complete ligand occupancy and

stabilization of the fully open channel. Further investigation will be necessary to reveal a conductive state of Kir7.1.

Collectively, these findings provide a framework to explain Kir7.1's ion selectivity, lipid sensitivity, and pharmacological modulation. They also offer a structural basis for the synergistic effects between steroids and phosphoinositides, proposing that full channel activation may require not only $PIP_2$ coordination but also displacement of inhibitory lipids by activating steroids. These insights may inspire therapeutic strategies targeting Kir7.1 in diseases where its regulation is disrupted.

## Methods
### Chemicals and solutions
All chemicals were purchased from Sigma-Aldrich, Anatrace, and Fisher Scientific. All the reagents used in this study were of the highest purity grade (BioXtra or similar). Steroids progesterone, pregnenolone, allo-pregnanolone, YX39 (5α-pregnane-3,20-dione), and 17OHPC were purchased from either Tocris Inc or Millipore. Remaining steroids were

synthesized by Dr. Covey lab as described in the methods. DMSO or ethanol was used as a solvent with final DMSO concentrations <0.01%.

## Molecular biology and virus production

Full-length human Kir7.1 (isoform A) was cloned from pIRES-Kir7.1[19,39] and engineered with TEV protease cleavage site, enhanced green fluorescent protein (EGFP), and 10 × His tag at C-terminus in both pIRES backbone and baculoviral backbone. Transient transfection was performed while establishing the BacMam system. To purify large amounts of plasmid DNA from liters of culture, either the PureLinkTM HiPure Expi Plasmid Gigaprep Kit (Invitrogen) or the GenElute™ HP Select Plasmid Gigaprep Kit was employed with transient transfection. The protocol for each kit was followed as recommended by the manufacturer. For the virus production, the BacMam system was used. The resulting baculoviral DNA construct was transformed into DH10Bac to generate the recombinant bacmid DNA, Bacmid containing DNA was then used to transfect Sf9 insect cells cultured in Sf-900™ II SFM (Gibco). The resulting baculovirus was amplified by infecting fresh Sf9 cells and incubating them at 27 °C on an orbital shaker for 72 h, followed by harvesting by centrifugation at 2800 × g.

## Protein expression

Docked state: Expi293F™ GnTI- mammalian cell culture were cultured in Expi293F™ Expression Medium (Thermo Fisher Scientific) and transient infected with amplified pIRES-Kir7.1-EGFP plasmids using linear polyethylenimine hydrochloride (MW 40 K) in a ratio of 3:1 PEI-HCl-40K to DNA at cell density 2.5 × 10^6 cells/mL. After 8 to 16 h, 10 mM sodium butyrate (NaBu) was added to enhance protein expression, and the culture was transferred to a 30 °C shaking incubator with 8% CO$_2$. Cells were harvested 60–72 h post-infection by centrifugation at 2800 × g for 20 min, washed with phosphate-buffered saline (PBS), pelleted again, and flash-frozen in liquid nitrogen for storage at −80 °C.

Extended state and compound bound state: For protein expression, Expi293F™ mammalian cell culture in mid-log phase were infected with amplified baculovirus at 10% (v/v) at cell density 2.5 × 10^6 cells/mL. After 8 to 14 h, 10 mM NaBu was added and the culture was transferred to a 30 °C shaking incubator with 8% CO$_2$. Cells were harvested 60–72 h post-infection by centrifugation at 2800 × g for 20 min, washed with PBS, pelleted again, and flash-frozen in liquid nitrogen for storage at −80 °C.

## Purification of Kir7.1 channels

Whole cell pellets expressing Kir7.1 were resuspended in hypotonic buffer (10 mM Tris-HCl pH 7.5, 15 mM KCl, 2 mM EDTA, 3 mM DTT, protease inhibitor cocktail, and DNase I) and rotated at 4 °C for 30 min. The cells were then homogenized by Dounce homogenizer and transferred into pre-chilled ultracentrifuge tubes for ultracentrifugation at 100,000 × g for 30 min at 4 °C. The resulting membrane pellet was resuspended in solubilization buffer (20 mM Tris-HCl pH 7.5, 150 mM KCl, 1 mM EDTA, 2 mM DTT, protease inhibitor cocktail, and 1% GDN) and incubated at 4 °C for 1 h with gentle rotation to extract membrane proteins. Another round of ultracentrifugation at 100,000 × g for 30 min was applied to remove any insoluble materials. To purify the solubilized Kir7.1 protein, washed GST-GFP nanobody resin was added to the supernatant at a ratio of 1 mL resin per 1000 mL cell, and the mixture was gently rotated at 4 °C for 1 h. The resin was then collected using a gravity flow column and washed sequentially with washing buffer 20 mM Tris-HCl pH 7.5, 300 mM KCl, 2 mM ATP, 2 mM MgCl$_2$) supplemented with stepwise decreasing concentration of GDN (0.5%, 0.25%, and 0.02%). Kir7.1 was eluted with 5 column volumes (CV) of elution buffer (20 mM Tris-HCl pH 7.5, 150 mM KCl, 0.02% GDN) by 1 h on-column cleavage with TEV protease. The eluted protein was concentrated and subjected to a size-exclusion chromatography (SEC) on a column pre-equilibrated with SEC buffer (20 mM Tris-HCl pH 7.1, 150 mM KCl, 0.02% GDN). Peak fractions were pooled and concentrated to a final protein concentration of 4 to 7 mg/ml for downstream structural studies.

## Cryo-electron microscopy sample preparation, data collection, and processing

For cryo-EM sample preparation, purified Kir7.1 was frozen directly on UltrAuFoil™ Au R0.6/1 300-mesh grids (Quantifoil). 1 mM ENT-17OHPC and 1 mM diC8-PIP$_2$ were added to purified proteins and incubated for 30 min before vitrification. Grids were glow-discharged for 45 s using Pelco easiGlow™ system, then plunge-frozen using a Vitrobot Mark IV (Thermo Fisher Scientific) set to 100% humidity and 4 °C temperature.

Three cryo-EM datasets were collected using 300 keV Titan Krios microscopes located at cryo-EM facility center in the Case Western Reserve University, the National Center for CryoEM Access and Training (NCCAT) and at the Electron Microscopy core facility at the University of Missouri-Columbia (MIZZOU).

**Extended state.** This dataset was collected on a 300 keV Titan Krios (Thermo Fisher Scientific) equipped with a Selectris energy filter and a Falcon 4 direct detector at MIZZOU cryo-EM Facility. A total of 12,677 movie series were collected with EPU software at a defocus range of −0.4 to −2.8 μm at a magnification of 130,000 × g, corresponding to a calibrated pixel size of 0.9525 Å. Movie frames were drift-corrected and dose-weighted using "Patch Motion Correction" in cryoSPARC[40]. Contrast transfer function (CTF) parameters were estimated using "Patch CTF Estimation" in cryoSPARC. A total of 3,859,111 particles were automatically picked using the blob picker. After four rounds of 2D classification, 293,791 particles from high-quality classes were selected to generate an ab initio model. These particles were further subjected to heterogeneous refinement into three classes, resulting in a "good" class containing 109,323 particles, which were subsequently refined using non-uniform refinement with C4 symmetry. Local refinement with TMD mask and CTD mask generated using Chimera were applied to improve local resolution. Final reconstructions reached 3.3 Å (overall) and 2.8 Å (TMD) resolution.

The docked and compound bound states were processing using a similar strategy as the extended state (Figs. S6 and S10 and Table 1). Docked state dataset was collected on 300 keV Titan Krios equipped with a BioQuantum Energy Filter (slit width 20 eV) (Gatan) and a K3 direct electron detector (Gatan) at the CWRU Cryo-EM Facility. A total of 5666 movie series were collected using SerialEM at a defocus range of −0.6 to −2.5 μm at a magnification of 105,000 × g, corresponding to a calibrated pixel size of 0.848 Å. Final reconstructions reached average 3.9 Å resolution. ENT-17OHPC bound state dataset was collected on 300 keV Titan Krios equipped with a Selectris energy filter and a Falcon 4 direct detector at the NCCAT Cryo-EM Facility. A total of 12,187 movie series were collected using EPU at a defocus range of −0.6 to −2.5 μm at a magnification of 165,000 × g, corresponding to a calibrated pixel size of 0.73 Å. Final reconstructions reached 3.4 Å (C4) and 4 Å (C1) resolution.

## Model building and refinement

Initial atomic models of extended Kir7.1, docked Kir7.1, and ENT-17OHPC bound Kir7.1 were generated by fitting AlphaFold 3[41] structure into their respective cryo-EM maps as rigid body using UCSF ChimeraX v1.8[42]. Interactive molecular dynamics flexible fitting was performed in ISoLDE[43], followed by manual model correction and rebuilding in Coot[44]. The PIP$_2$, cholesterol, and ENT-17OHPC molecules, and their geometry, were generated from SMILE[45] codes using eLBOW[46]. Subsequent real-space refinement was carried out using Phenix.real_space_refine, employing secondary structure, Ramachandran, and rotamer restraints. The quality of the refined model was assessed by MolProbity[47], shown in Table 1.

## Whole-cell electrophysiology

Recordings from heterologously expressed human Kir7.1 was done as reported previously[16]. Specifically, HEK293T cells were transfected with full-length human Kir7.1 (isoform A) cloned into pIRES-EGFP as reported in refs. 19,39, as well as with full-length human Kir7.1 engineered

with TEV protease cleavage site and EGFP. There were no difference in Kir7.1 currents generated by either constructs. For experiments involving cholesterol competition, a stable transfected cell line HEK293T-hKir7.1 was generated by VectorBuilder, Inc (Chicago, IL). Specifically, HEK293T cells were modified by lentiviral transfection with full-length human *KCNJ13* and dTomato using generated vector: pLV-EF1A-hKCNJ13-CMV-dTomato(ns):P2A-Puro. Individual clones were selected and tested using qPCR and electrophysiology, after which positive clone was amplified and used in the mentioned above experiments. Current recording were conducted at ambient temperature using voltage-clamp protocol. Patch pipettes (resistance 8–10 MΩ) were filled with a cesium-based internal solution containing (in mM): 130 CsMeSO₃, 20 HEPES, 5 BAPTA, and 1 MgCl₂ (pH 7.3, adjusted with CsOH; 295 mOsm/kg). Seals were formed in Krebs solution containing (in mM): 135 NaCl, 5 KCl, 1 CaCl₂, 1 MgSO₄, KH₂PO₄, 20 HEPES, and 5.5 D-glucose (pH 7.3, adjusted with KOH; 310 mOsm/kg). Break-in was achieved by light suction, after which cells were stimulated every 5 s using voltage ramps from −80 to +80 mV with a holding potential of 0 mV. Access resistance ranged from 10 to 25 MΩ, and membrane capacitance for HEK293T cells was recorded to serve as a proxy for cell surface area. All steroid compounds were applied via external (bath) solution contained (in mM): 130 KMeSO₃, 1 MgCl₂, 43 HEPES, pH 7.3, adjusted with KOH; 310 mOsm/kg. All buffers, including those containing steroids, were applied under continuous perfusion. Where indicated, the Kir7.1 antagonist ML418 (10 µM; Tocris, Inc cat no: 6889) and compounds were co-applied directly to the bath solution. Phosphoinositide diC8-PI₄,₅P₂ (cat#P4508) was purchased from Echelon Biosciences, Inc (Salt Lake City, UT) and dissolved directly in the pipette solution. Polyphosphoinositide-Binding Peptide, PBP10 (PBP) was purchased from MilliporeSigma (CAS No.: 794466-45-8) and dissolved in DMSO. PBP was applied via bath solution. Data were acquired using Clampex 10.5 software (Molecular Devices), controlled via an AXO-PATCH 200B amplifier and a Digidata 1550 A digitizer (Molecular Devices), integrated with a Humbug noise eliminator. Current amplitudes were normalized to membrane capacitance to calculate current density. Capacitance artifacts were removed from current traces during analysis using OriginPro 8.6 (OriginLab).

### Statistics and reproducibility

Electrophysiological data were analyzed and visualized using Clampfit (pClamp 10.3, Molecular Devices, Sunnyvale, CA, USA) and OriginPro 9.6 (OriginLab, Northampton, MA, USA). Curve fitting and quantitative measurements were performed in these platforms. All data are presented as mean ± standard error of the mean (S.E.M.), and n corresponds to the number of individual cells analyzed as biological replicates. For all experiments, sample sizes were chosen based on standards commonly used in the field and on our prior experience with similar assays, which indicated that these numbers are sufficient to detect biologically meaningful effects. Current densities were calculated by normalizing the amplitude of the current to the cell capacitance (Cm, pF). One-way analysis of variance and post hoc Tukey's test were used for comparisons of mean values and statistical significance. No data were excluded from the analyses except when capacitance artifacts were graphically removed.

### Chemical synthesis.

The synthesis of the following compounds was reported previously: *ent*-progesterone[48], *ent*-DHEA[49], *ent*-pregnenolone[49], *ent*-allopregnanolone[50]. These procedures are also described below.

*ent*-progesterone[48]. Borane–tetrahydrofuran (THF) complex (2 mL, 2 mmol) was added to a stirred solution of *ent*-[(17Z)-Pregna-5,17(20)-dien-3-one cyclic (1,2-ethanediyl acetal)] (160 mg, 0.469 mmol) in dry THF (10 mL) under a nitrogen atmosphere at room temperature, and the reaction was continued for 5 h. Aqueous sodium hydroxide (5 mL, 10%) and aqueous hydrogen peroxide (5 mL, 30%) were then

added, and stirring was continued for 2 h. The reaction solution was then poured into aqueous saturated sodium chloride and extracted with ethyl acetate. The ethyl acetate was dried over anhydrous sodium sulfate, and the crude product obtained on removal of solvent was used in the next reaction. Celite 545 (0.5 × g) and pyridinium chlorochromate (600 mg, 2.78 mmol) were added to crude product (200 mg) from the previous reaction dissolved in dichloromethane (20 mL) and the reaction mixture was stirred at room temperature for 3 h. Then the reaction mixture was poured onto a silica gel column, and the product was eluted from the column with ethyl acetate. After solvent removal, the uncharacterized oxidation product was dissolved in acetone, treated with dilute hydrochloric acid (6 mL, 30%), and stirred at room temperature for 0.5 h. The acetone was then removed, and the residue in the remaining water was isolated by extraction with ethyl acetate (50 mL). The ethyl acetate was washed with 5% aqueous sodium bicarbonate (50 mL) and then with aqueous saturated sodium chloride (50 ml) and dried over anhydrous sodium sulfate. After solvent removal, the crude *ent*-progesterone product was purified by silica gel column chromatography using 30% ethyl acetate in hexanes as eluent. The purified *ent*-progesterone was recrystallized from hexanes to obtain a white crystalline solid (120 mg, 82%) that had mp 120–121 °C; [α]²⁵_D −200 (c 0.25, CHCl₃); IR 2939, 2854, 1704, 1674, 1615, 1448, 1435, 1355, 1227, 1190, 1162, 950, 864 cm⁻¹; NMR (CDCl₃) δ 5.73 (s, 1H, C = CH), 2.12 (s, 3H, COCH₃), 1.19 (s, 3H, CH₃), 0.67 (s, 3H, CH₃); (CDCl₃) NMR δ 209.42 (C-20), 199.54 (C-3), 171.01 (C-5), 123.95 (C-4), 63.40, 55.92, 53.53, 43.79, 38.52, 38.43, 35.58, 35.40, 33.80, 32.62, 31.74, 31.35, 24.20, 22.67, 20.84, 17.19, 13.15. *Anal.* Calculated (Calcd) for C₂₁H₃₀O₂: C, 80.20, H, 9.61. Found: C, 80.40; H, 9.59.

*ent*-DHEA[49]. (3α,8α,9ß,10α,13α,14ß)-3-Hydroxyandrost-5-en-17-one. (8α,9ß,10α,13α,14ß)-Androst-4-ene-3,17-dione (2.68 × g, 9.37 mmol) was mixed with *t*-BuOK (10.5 × g, 93.8 mmol) in a flask equipped with a rubber septum and purged with N₂ for 25 min. *t*-BuOH (82 mL) was slowly added by syringe, and the stirred reaction was allowed to proceed for 3 h. The reaction was quenched by the rapid addition of the reaction mixture to 10% aqueous acetic acid (282 mL). After stirring for 10 min, the acetic acid was neutralized by the addition of solid NaHCO₃, and the mixture was extracted with EtOAc (3 × 150 mL). The combined extracts were dried, filtered, and removed to give crude (8α,9ß,10α,13α,14ß)-Androst-5-ene-3,17-dione (3.26 × g) as a yellow solid which was immediately dissolved in THF (57 mL), added to a flask equipped with a rubber septum, cooled to −78 °C in a dry ice/acetone bath, and purged with N₂ for 10 min. LiAl(*t*-BuO)₃H (15 mL of a 1.0 M solution in THF) was added by syringe. After 3.5 h, additional LiAl(*t*-BuO)₃H (8 mL of a 1.0 M solution in THF) was added, and 40 min later the mixture was poured into chilled 1 N HCl (440 mL). The mixture was extracted with EtOAc (3 × 150 mL), the combined organic extracts were washed with saturated NaHCO3 (2 × 150 mL), dried, and filtered, and the solvents were removed to afford a yellow solid (3.58 × g). 1H NMR analysis revealed a mixture of steroids. This steroid mixture was recycled following the same procedure, and the resulting yellowish solid was purified by flash column chromatography to give a white solid (1.64 × g). Recrystallization from EtOAc and hexanes gave *ent*-DHEA (1.32 × g, 49%) as a white solid: mp 139–140 °C; [α]²³_D −4.68 (c 1.58, CHCl₃); ee 98.9%; IR 3422, 2933, 1738, 1620, 1454, 1375, 1058, 1029 cm⁻¹; ¹H NMR (CDCl₃) δ 5.37 (m, 1H, C = CH), 4.09 (m, 1H, C*H*OH), 1.02 (s, 3H, CH₃), 0.87 (s, 3H, CH₃); ¹³C NMR (CDCl₃) δ 221.49, 141.10, 120.95, 71.51, 51.67, 50.11, 47.45, 42.08, 37.06, 36.51, 35.74, 31.43, 31.35, 31.29, 30.64, 21.73, 20.21, 19.27, 13.38. *Anal.* Calcd for (C₁₉H₂₈O₂): C,79.12; H, 9.78. Found: C, 78.88; H, 9.72.

*ent*-pregnenolone[49]. 3α,8α,9ß,10α,13α,14ß,20α)-3-[[(1,1-Dimethylethyl)di-methylsilyl]oxy]pregn-5-en-20-one (0.66 × g, 1.53 mmol) was dissolved in stirred THF (9 mL), and (n-Bu)₄NF (9 mL of a 1 M solution in THF) was added. The following morning (~20 h), saturated NH₄Cl (40 mL) was added and the reaction mixture was extracted with CH₂Cl₂ (3 × 30 mL). The combined organic extracts were dried and filtered,

and the solvents were removed to afford a yellow oil. Flash column chromatography (silica gel eluted with $CH_2Cl_2$ then 20% EtOAc in hexanes) gave a white solid (490 mg) which was recrystallized from EtOAc and hexanes to give *ent*-pregnenolone as flaky white crystals (450 mg, 93%): mp 188–191 °C; $[\alpha]^{28}_D$ −26.0 (c 1.00, EtOH); ee 99.4%; D IR 3434, 2929, 2885, 1699, 1434, 1360, 1315, 1234, 1195, 1060 cm$^{-1}$; $^1$H NMR δ 5.34 (m, 1H, C=CH), 3.51 (m, 1H, C*H*OH), 2.52 (t, *J* = 9 Hz, 1H), 2.11 (s, 3H, CH$_3$), 1.00 (s, 3H, CH$_3$), 0.62 (s, 3H, CH$_3$); $^{13}$C NMR δ 209.7, 140.9, 121.4, 71.64, 63.7, 56.9, 49.9, 43.9, 42.2, 38.8, 37.2, 36.4, 31.8, 31.7, 31.5, 31.4, 24.4, 22.7, 21.0, 19.2, 13.1. *Anal*. Calcd for $C_{21}H_{32}O_2$: C, 79.70, H, 10.19. Found: C, 79.60; H, 10.11.

*ent*-allopregnanolone[50]. Reduction of the enone system in (8α,9β,10α,13α,14β,17α)-17-hydroxyandrost-5-en-3-one using Li–NH$_3$(l) gave (5β,8α,9β,10α,13α,14β,17α)-17-hydroxyandrostan-3-one (82%), and Jones oxidation of (5β,8α,9β,10α,13α,14β,17α)-17-hydroxyandrostan-3-one gave (5β,8α,9β,10α,13α,14β)-androstane-3,17-dione (91%). Selective reduction of the C-3 carbonyl group of (5β,8α,9β,10α,13α,14β)-androstane-3,17-dione using K-Selectride® in THF at −78 °C gave (3β,5β,8α,9β,10α,13α,14β)-3-hydroxyandrostan-17-one (86%), and this compound was then acetylated using pyridine–(Ac)$_2$O to obtain acetoxy (3β,5β,8α,9β,10α,13α,14β)-3-acetyloxyandrostan-17-one (94%). (3β,5β,8α,9β,10α,13α,14β)-3-acetyloxyandrostan-17-one was converted into a diastereomeric mixture of compounds (3β,5β,8α,9β,10α,13α,14β,17α)-3-acetyloxyandrostan-17-carbonitrile and (3β,5β,8α,9β,10α,13α,14β,17β)-3-acetyloxyandrostan-17-carbonitrile (82%) by a two step procedure which involves the use of LiCN and diethyl cyanophosphonate to produce intermediate cyanophosphonate diastereomers, and the subsequent reduction of the cyanophosphonates using SmI$_2$–THF. (3β,5β,8α,9β,10α,13α,14β,17α)-3-acetyloxyandrostan-17-carbonitrile and (3β,5β,8α,9β,10α,13α,14β,17β)-3-acetyloxyandrostan-17-carbonitrile were converted into (3β,5β,8α,9β,10α,13α,14β,17α)-3-hydroxypregnan-20-one and (3β,5β,8α,9β,10α,13α,14β,17β)-3-hydroxypregnan-20-one using CH$_3$MgBr in THF at reflux. After HPLC separation, (3β,5β,8α,9β,10α,13α,14β,17α)-3-hydroxypregnan-20-one (*ent*-allopregnanolone) and (3β,5β,8α,9β,10α,13α,14β,17β)-3-hydroxypregnan-20-one were obtained in yields of 45 and 17%, respectively. The *ent*-allopregnanolone was obtained as white crystals, mp 170–172 °C (from EtOEt–EtOAc); $[\alpha]^{24}_D$ 98.4 (CHCl$_3$; 97% ee); IR 3382, 2930, 2871, 1707, 1447, 1356, 1154, 1004 cm$^{-1}$; $^1$H NMR (CDCl$_3$0 δ 4.04 (1H, s, C*H*OH), 2.53 (1H, t, *J* = 9.5 Hz, C*H*COCH$_3$), 2.11 (3H, s, COCH$_3$), 0.77 (3H, s, CH$_3$), 0.60 (3H, s, CH$_3$); $^{13}$C NMR (CDCl$_3$) δ 209.80, 66.46, 13.45, 11.15, 63.80, 56.74, 54.15, 44.25, 39.05, 36.07, 35.80, 35.44, 32.15, 31.91, 31.54, 28.95, 28.40, 24.34, 22.72, 20.76. *Anal*. Calcd for $C_{21}H_{34}O_2$: C, 79.2; H, 10.8. Found: C, 79.3; H, 10.6.

Remaining compounds, i.e. *ent*-17OHC, MQ331, MQ344, MQ345 and MQ346, MQ351 and YX54 were synthesized as described below.

a) Procedures for the synthesis of *ent*-17OHPC in six steps.

*ent*-Androstenedione (step 1). To a solution of *ent*-testosterone[51] (250 mg, 0.87 mmol) in $CH_2Cl_2$ (10 mL) was added Dess-Martin periodinane (550 mg, 1.3 mmol) at 23 °C. After 1.5 h, saturated aqueous

NaHCO$_3$ was added, and the product was extracted into $CH_2Cl_2$ (20 mL × 3). The combined extracts were dried over anhydrous Na$_2$SO$_4$ and filtered. The solvent was removed and the residue was purified by flash column chromatography (silica gel eluted with 20% EtOAc in hexanes) to give *ent*-androstenedione (**1**, 246 mg, 99%): $^1$H NMR (400 MHz, CDCl$_3$) δ 5.70 (s, 1H), 2.44–0.88 (m, 19H), 1.17 (s, 3H), 0.87 (s, 3H); $^{13}$C NMR (100 MHz, CDCl$_3$) δ 220.1, 199.1, 170.2, 123.9, 53.6, 50.7, 47.3, 38.5, 35.6, 35.5, 35.0, 33.8, 32.4, 31.1, 30.6, 21.6, 20.1, 17.2, 13.5.

**ent*-[(17α)-17-Hydroxy-3-oxoandrost-4-ene-17-carbonitrile] (step 2)**. Acetic acid (0.1 mL, 1.4 mmol) was added over a period of 20 min at 23 °C to *ent*-androstenedione (246 mg, 0.86 mmol) and KCN (227 mg, 3.5 mmol) in MeOH (5 mL). After 16 h, acetic acid (0.15 mL, 2.1 mmol) was added and the reaction was stirred for 15 min. After dilution with water and saturated aqueous NaHCO$_3$, the product was extracted into EtOAc (30 mL × 2). The combined extracts were washed with brine (50 mL), water (50 mL), dried over anhydrous Na$_2$SO$_4$ and filtered. The solvent was removed to give *ent*-steroid **2** (270 mg, 100%): $^1$H NMR (400 MHz, CDCl$_3$) δ 5.75 (s, 1H), 3.50–3.48 (m, 1H), 2.48–0.86 (m, 19H), 1.20 (s, 3H), 0.98 (s, 3H); $^{13}$C NMR (100 MHz, CDCl$_3$) δ 200.1, 171.3, 123.9, 120.9, 77.6, 53.0, 49.0, 47.6, 38.5, 38.1, 35.9, 35.6, 33.8, 32.7, 31.9, 29.3, 23.8, 20.3, 17.3, 16.2.

**ent*-[[(17α)-3,3-[1,2-Ethanediylbis(oxy)]-17-hydroxyandrost-5-ene-17-carbonitrile] (step 3)**. To a solution of *ent*-steroid **2** (270 mg, 0.86 mmol) in benzene (80 mL) was added ethyl glycol (2 mL, 35.8 mmol) and *p*-toluenesulfonic acid (20 mg) at 23 °C. The reaction was refluxed in a flask equipped with a Dean-Stark apparatus. After 16 h, the reaction was cooled to 23 °C and saturated aqueous NaHCO$_3$ was added. The product was extracted into EtOAc (100 mL). The EtOAc was washed with brine (100 mL × 5), dried over anhydrous Na$_2$SO$_4$ and filtered. The solvent was removed and the residue was purified by flash column chromatography (silica gel eluted with 10–20% EtOAc in hexanes) to give *ent*-steroid **3** (220 mg, 72%): δ 5.32-5.31 (s, 1H), 4.04–3.95 (m, 4H), 3.38 (s, 1H), 2.61–2.57 (m, 1H), 2.44–2.37 (m, 1H), 2.13–0.88 (m, 17H), 1.04 (s, 3H), 0.95 (s, 3H); $^{13}$C NMR (100 MHz, CDCl$_3$) δ 139.9, 121.9, 121.4, 109.7, 77.6, 64.3, 64.3, 49.0, 48.9, 48.7, 41.9, 38.4, 36.4, 35.9, 32.2, 31.7, 30.8, 29.5, 24.1, 20.4, 19.0, 16.1.

**ent*-[(17α)-3,3-[1,2-ethanediylbis(oxy)]-17-(1-ethoxyethoxy)-androst-5-ene-17-carbonitrile] (step 4)**. To a solution of *ent*-steroid **3** (220 mg, 0.62 mmol) in $CH_2Cl_2$ (5 mL) ethyl vinyl ether (1 mL, 10.5 mmol) and pyridine hydrochloride (10 mg, 0.09 mmol) were added at 23 °C. The reaction was heated at 55 °C for 16 h. After cooling to 23 °C, the solvent was removed and the residue was purified by flash column chromatography (silica gel eluted with 10–20% EtOAc in hexanes) to give *ent*-steroid **4** (230 mg, 85%): $^1$H NMR (400 MHz, CDCl$_3$) δ 5.34–5.33 (m, 1H), 5.09–5.05 (m, 1H), 4.00–3.90 (m, 4H), 3.59–3.55 (m, 1H), 3.49–3.45 (m, 1H), 2.58–2.50 (m, 2H), 2.22–0.88 (m, 23 H), 1.04 (s, 3H), 0.96 (s, 3H); $^{13}$C NMR (100 MHz, CDCl$_3$) δ 140.2, 121.5, 119.3, 109.3, 98.3, 83.3, 64.4, 64.2, 59.5, 50.0, 49.2, 49.1, 41.7, 36.5, 36.3, 34.4, 32.4, 31.7, 31.0, 29.4, 24.3, 20.4, 20.4, 18.9, 16.2, 15.3.

**ent*-(17α-Hydroxyprogesterone) (step 5)**. A solution of *ent*-steroid **4** (230 mg, 0.53 mmol) in Et$_2$O (10 mL) was cooled to 0 °C and MeLi (3 mL, 1.6 M in Et$_2$O, 4.8 mmol) was added. The reaction was heated at 40 °C for 5 h. The reaction was cooled to 0 °C and Et$_2$O (20 mL) was added, followed by the addition of 3N HCl (5 mL, 15 mmol). Stirring was continued for 16 h and the product was extracted into EtOAc (100 mL × 2). The combined extracts were washed with saturated aqueous NaHCO$_3$ (50 mL), water (50 mL), brine (50 mL), dried over Na$_2$SO$_4$ and filtered. The solvent was removed and the residue was purified by flash column chromatography (silica gel eluted with 10–20% EtOAc in hexanes) to give *ent*-[17α-hydroxyprogesterone] (**5**, 110 mg, 62%): $^1$H NMR (400 MHz, CDCl$_3$) δ 5.67 (s, 1H), 3.24 (s, 1H), 2.64–2.61 (m, 1H), 2.21 (s, 3H), 1.14 (s, 3H), 0.68 (s, 3H), 2.36–0.63 (m, 18H); $^{13}$C NMR (100 MHz, CDCl$_3$) δ 211.4, 199.5, 171.3, 123.7, 89.7, 53.2, 49.9, 47.7, 38.4, 35.5, 35.3, 33.7, 33.2, 32.7, 31.9, 30.0, 27.5, 23.7, 20.4, 17.2, 15.1.

***ent*-[17α-Hydroxyprogesterone caproate] (step 6, *ent*-17-OHPC).** *ent*-[17α-Hydroxyprogesterone] (**5**, 110 mg, 0.33 mmol) and hexanoic anhydride (140 mg, 0.7mmol), *p*-toluenesulfonic acid (10 mg) and dry benzene (5 mL) were heated at 80 °C until a clear solution was obtained. The solution was allowed to stand at room temperature for 18 h, poured into ice and water, and stirred to effect hydrolysis of the excess anhydride. The product was extracted into Et$_2$O and the combined extracts were washed with NaOH solution (50 mL, 1M), water (50 mL), dried over anhydrous Na$_2$SO$_4$ and filtered. The solvent was removed, and the residue was purified by flash column chromatography (silica gel eluted with 20% EtOAc in hexanes) to give *ent*-[17α-Hydroxyprogesterone caproate] (**6**, 85 mg, 60%): $^1$H NMR (400 MHz, CDCl$_3$) δ 5.74 (s, 1H), 2.98–2.90 (m, 1H), 2.50–0.87 (m, 29H), 2.03 (s, 3H), 1.19 (s, 3H), 0.67 (s, 3H); $^{13}$C NMR (100 MHz, CDCl$_3$) δ 204.0, 199.3, 173.3, 170.6, 123.9, 96.4, 53.1, 51.2, 46.8, 38.5, 35.7, 35.6, 34.4, 33.9, 32.7, 31.9, 31.2, 31.0, 30.3, 26.3, 24.5, 23.8, 22.2, 20.6, 17.3, 14.3, 13.9.

High-resolution mass spectral (HRMS) were (m/z): [M+H]+ calculated for C$_{27}$H$_{41}$O$_4$, 429.3005; found, 429.2999. Optical rotation: $\alpha_D^{23}$ -63 (CHCl$_3$, *c* = 0.20). Reported literature value for natural 17α-Hydroxyprogesterone caproate: $\alpha_D^{25}$ + 61 (CHCl$_3$, *c* =1)[52].

b) Procedures for the synthesis of **MQ331** in four steps.

**17β-Hydroxyandrost-5-en-3-one, cyclic 1,2-ethanediyl acetal (step 1).** To a solution of testosterone (**1**, 600 mg, 2.08 mmol) in toluene (150 mL) was added ethylene glycol (0.5 mL) and *p*-toluenesulfonic acid (20 mg) at 23 °C. The reaction was refluxed in a flask equipped with a Dean–Stark apparatus for 16 h. After cooling to 23 °C, solid NaHCO$_3$ (200 mg) was added and stirring was continued for 10 min. Aqueous NaHCO$_3$ added and the product was extracted into EtOAc (300 mL). The combined organic layers were washed with brine (100 mL x 4), dried over anhydrous Na$_2$SO$_4$, filtered and the solvent was removed. The residue was purified by flash column chromatography (silica gel, eluted with 30% EtOAc in hexanes) to give steroid **2** (520 mg, 75%): $^1$H NMR (400 MHz, CDCl$_3$) δ 5.31–5.30 (m, 1H), 4.09–3.87 (m, 4H), 3.62 (*t*, *J* = 8.6 Hz, 1H), 2.55–2.51 (m, 1H), 2.10–0.88 (m, 19H), 1.00 (s, 3H), 0.72 (s, 3H); $^{13}$C NMR (100 MHz, CDCl$_3$) δ 140.0, 121.7, 109.3, 81.6, 64.3, 64.1, 51.2, 49.6, 42.6, 41.6, 36.5, 36.4, 36.2, 31.8, 31.2, 30.9, 30.3, 23.3, 20.5, 18.8, 10.9.

**17β-[[4-[3-(Trifluoromethyl)-3*H*-diazirin-3-yl]phenyl]methoxy]-androst-5-en-3-one, cyclic 1,2-ethanediyl acetal (step 2).** To a solution of steroid **2** (270 mg, 0.81 mmol) in THF (10 mL was added KH in mineral oil (267 mg, 2 mmol) at 23 °C and the reaction was stirred for 30 min. 3-(4-(Iodomethyl)phenyl)-3-(trifluoromethyl)-3*H*-diazirine (531 mg, 1.6 mmol) in THF (4 mL) was added and the reaction was heated to 50 °C for 16 h. After cooling to 23 °C, water was added and the product was extracted into EtOAc (15 mL × 2). The combined organic layers were dried over anhydrous Na$_2$SO$_4$, filtered and the solvent was removed. The residue was purified by flash column

chromatography (silica gel, eluted with 10% EtOAc in hexanes) to give steroid **3** (145 mg, 34%): $^1$H NMR (400 MHz, CDCl$_3$) δ 7.39 (d, *J* = 8.2 Hz, 2H), 7.18 (d, *J* = 8.2 Hz, 2H), 5.36-5.35 (m, 1H), 4.55 (s, 2H), 4.01–3.86 (m, 4H), 3.43 (t, *J* = 8.2 Hz, 1H), 2.61–2.57 (m, 1H), 2.44–0.88 (m, 18H), 1.05 (s, 3H), 0.85 s, 3H); $^{13}$C NMR (100 MHz, CDCl$_3$) δ 141.2, 127.9, 127.4 (2 × C), 126.4 (2 × C), 126.3, 121.8, 109.4, 88.6, 70.8, 64.4, 64.2, 51.4, 49.7, 42.8, 41.7, 37.7, 36.6, 36.2, 31.7, 31.3, 31.0, 27.8, 23.4, 20.7, 20.6, 18.8, 11.6; HRMS (m/z): [M+H]+ calculated for C$_{28}$H$_{34}$F$_3$N$_2$O$_2$, 487.2567; found, 487.2566.

**17β-[[4-[3-(Trifluoromethyl)-3*H*-diazirin-3-yl]phenyl]methoxy]-androst-4-en-3-one (step 3, MQ331).** To a solution of steroid **3** (145 mg, 0.27 mmol) in acetone (30 mL) was added *p*-toluenesulfonic acid (50 mg) at 23 °C. After 3 h, acetone was removed under reduced pressure and the residue was purified by flash column chromatography (silica gel, eluted with 30% EtOAc in hexanes) to give steroid 4 (**MQ331**, 145 mg, 34%): $^1$H NMR (400 MHz, CDCl$_3$) δ 7.39 (d, *J* = 8.2 Hz, 2H), 7.17 (d, *J* = 8.2 Hz, 2H), 5.73 (s, 1H), 4.54 (s, 2H), 3.41 (t, *J* = 8.2 Hz, 1H), 2.43–0.89 (m, 19H), 1.05 (s, 3H), 0.85 s, 3H); $^{13}$C NMR (100 MHz, CDCl$_3$) δ 199.5, 171.2, 141.1, 127.4 (3 × C), 126.4 (2 × C), 123.8, 88.3, 70.7, 53.8, 50.6, 42.9, 38.6, 37.5, 35.7, 35.3, 33.9, 32.7, 31.5, 27.8, 23.2, 20.6, 17.3 (2 × C), 11.7.

c) Procedures for synthesis of **MQ344, MQ345** (four steps) **and MQ346** in one step.

**3β-Hydroxy-17-(hydroxymethyl)-pregn-5-en-20-one (steps 1–2).** To a solution of the 3,20-diacetoxypregna-5,17(20)-diene[53] (**1**, 3 g, 7.5 mmol) in diethyl ether (200 mL) was added MeLi (1.6 M, 23.4 mL, 37.5 mmol) at −10 °C. After 1 h, dried ZnCl$_2$ in THF (20 mL) was added to the reaction and stirred for 1 h. Paraformaldehyde was heated up to 150–160 °C and the mono formaldehyde was introduced to reaction mixture by cannula and stirred for 16 h at 23 °C. Aqueous NH$_4$Cl was added and the product was extracted into EtOAc (200 mL × 2). The combined extracts were dried over anhydrous Na$_2$SO$_4$, filtered and the solvent removed. The residue was purified by flash column chromatography (silica gel, eluted with 50% EtOAc in hexanes) to give steroid **2** (540 mg, 21%): $^1$H NMR (400 MHz, CDCl$_3$) δ 5.27 (s, 1H), 4.10 (d, *J* = 10.1 Hz, 1H), 3.52 (d, *J* = 10.1 Hz, 1H), 3.45–3.38 (m, 1H), 2.48–0.78 (m, 21H), 2.16 (s, 3H), 0.94 (s, 3H), 0.61 (s, 3H); $^{13}$C NMR (100 MHz, CDCl$_3$) δ 212.9, 140.6, 120.9, 70.9, 67.1, 65.1, 53.1, 49.5, 45.0, 41.5, 37.0, 36.3, 32.5, 31.8, 31.7, 30.9, 28.8, 25.5, 24.4, 20.5, 19.0, 15.5.

**(3β-Hydroxy-pregn-5-en-20-one-17-yl)methyl hexanoate (step 3).** To a solution of steroid **2** (540 mg, 1.56 mmol) in pyridine (15 mL) was added hexanoic acid anhydride (0.4 mL, 1.64 mmol) at 0 °C. The reaction mixture was stirred at 0 °C for 16 h. Pyridine was removed under reduced pressure, and the residue was purified by flash column

chromatography (silica gel, eluted with 40% EtOAc in hexanes) to give steroid **4** (280 mg, 40%): ¹H NMR (400 MHz, CDCl₃) δ 5.28–5.27 (m,1H), 4.43 (d, $J$ = 9.9 Hz, 1H), 4.16 (d, $J$ = 9.9 Hz, 1H), 3.56–3.43 (m, 1H), 2.54–0.79 (m, 31H), 2.11 (s, 3H), 0.94 (s, 3H), 0.63 (s, 3H); ¹³C NMR (100 MHz, CDCl₃) δ 209.5, 174.0, 140.6, 121.0, 71.3, 67.9, 64.5, 53.1, 49.5, 45.5, 41.9, 37.1, 36.3, 34.0, 32.7, 32.0, 31.7, 31.2, 31.1, 28.7, 26.2, 24.4, 24.0, 22.1, 20.6, 19.2, 15.5, 13.7.

**(Pregn-4-ene-3,20-dione-17-yl)methyl hexanoate (step 4, MQ344) and (Pregn-4-ene-3,6,20-trione-17-yl)methyl hexanoate (step 4, MQ345).** To a solution of steroid **3** (190 mg, 0.43 mmol) in acetone at 0 °C was added Jones reagent until a brown color persisted. After 5 min, 2-propanol was added to consume excess Jones reagent. Water was added and the two steroid products were extracted into EtOAc (150 mL × 2). The combined extracts were dried over anhydrous Na₂SO₄, filtered and the solvent removed. The residue was purified by flash column chromatography (silica gel, eluted with 40% EtOAc in hexanes) to give **4, MQ344** (27 mg, 14%) and **5, MQ345** (72 mg, 37%).

**MQ344** had: ¹H NMR (400 MHz, CDCl₃) δ 5.74 (s, 1H), 4.47 (d, $J$ = 9.9 Hz, 1H), 4.22 (d, $J$ = 9.9 Hz, 1H), 2.58–0.87 (m, 30H), 2.17 (s, 3H), 1.19 (s, 3H), 0.73 (s, 3H); ¹³C NMR (100 MHz, CDCl₃) δ 209.3, 199.3, 174.0, 170.5, 124.0, 67.7, 64.4, 53.2, 52.3, 45.5, 38.5, 35.7, 35.6, 34.2, 33.9, 32.7, 32.6, 31.9, 31.2, 28.8, 26.3, 24.5, 24.4, 22.2, 20.6, 17.3, 15.8, 13.8; HRMS (m/z): [M+H]+ calculated for C₂₈H₄₃O₄, 443.3155; found, 443.3155

**MQ345** had: ¹H NMR (400 MHz, CDCl₃) δ 6.21 (s, 1H), 4.40 (d, $J$ = 9.9, 1H), 4.21 (d, $J$ = 9.9, 1H), 2.74–0.88 (m, 28H), 2.18 (s, 3H), 1.17 (s, 3H), 0.75 (s, 3H); ¹³C NMR (100 MHz, CDCl₃) δ 209.2, 201.3, 199.1, 173.9, 160.2, 125.8, 67.6, 64.1, 52.9, 50.3, 46.5, 45.6, 39.5, 35.5, 34.1, 34.0, 33.8, 32.3, 31.2, 29.1, 26.3, 24.5, 24.2, 22.2, 20.5, 17.5, 15.7, 13.8; HRMS (m/z): [M+H]+ calculated for C₂₈H₄₁O₅, 457.2949, found, 457.2951

**MQ346: Pregn-4-ene-3,20-dione-17α-yl) 2-(3-(but-3-yn-1-yl)-3H-diazirin-3-yl)acetate.** To a solution of 2-(3-(but-3-yn-1-yl)-3H-diazirin-3-yl)acetic anhydride (290 mg, 1.0 mmol) in benzene (10 mL) was added 17α-hydroxylprogesterone (165 mg, 0.5 mmol) and p-toluene-sulfonic acid (30 mg) at 23 °C. The reaction was heated to 65 °C for 4 h. The solvent was removed and the residue was purified by flash column chromatography (silica gel, eluted with 35% EtOAc in hexanes) to give **MQ346** (17 mg, 7%): ¹H NMR (400 MHz, CDCl₃) δ 5.74 (s,1H), 2.73–0.83 (m, 24H), 2.43 (s, 2H), 2.29 (s, 3H), 1.02 (s, 3H), 0.77 (s, 3H); ¹³C NMR (100 MHz, CDCl₃) δ 224.4, 211.8, 174.1, 139.1, 124.2, 89.9, 82.4, 69.5, 50.8, 48.4, 47.4, 39.7, 34.9, 33.7, 33.6, 32.0, 31.8, 31.6, 30.0, 29.7, 28.0, 25.4, 24.7, 24.1, 20.6, 18.8, 15.5, 13.2; LC-MS (ESI) m/z [M + Na]+ = 487.3; HRMS (m/z): [M+H]+ calculated for C₂₈H₃₇N₂O₄, 465.2748; found: 465.2803.

d) Procedures for synthesis of **MQ351** in two steps.

**3β-Hydroxy-17α-methyl-pregn-5-en-20-one (step 1).** Lithium wire was dissolved in stirred liquid ammonia (40 mL) and cooled in a dry ice-acetone bath to −78 °C. A solution of 16-Dehydropregnenolone acetate (**1**, 1.0 g, 2.81 mmol, purchased from Aaron Chemicals) in THF (40 mL) was first added dropwise followed by the addition of methyl iodide (3 mL) in 15 mL of anhydrous diethyl ether (15 mL). The blue color of the reaction disappeared during these additions. The cooling bath was removed and liquid ammonia allowed to evaporate overnight. Aqueous NH₄Cl was

added, and the product was extracted into EtOAc (150 mL × 2). The combined extracts were dried over anhydrous Na₂SO₄, filtered, solvent removed and the residue was purified by flash column chromatography (silica gel, eluted with 10–25% EtOAc in hexanes) to give steroid **2** (580 mg, 63%): ¹H NMR (400 MHz, CDCl₃) δ 5.33–5.32 (m, 1H), 3.52–3.46 (m, 1H), 2.64–2.56 (m, 1H), 2.30–0.89 (m, 19H), 2.10 (s, 3H), 1.11 (s, 3H), 0.98 (s, 3H), 0.64 (s, 3H); ¹³C NMR (100 MHz, CDCl₃) δ 212.6, 140.7, 121.2, 71.5, 61.5, 51.3, 49.6, 44.2, 42.1, 37.1, 36.4, 32.7, 32.0, 31.8, 31.4, 31.0, 27.8, 23.8, 21.5, 20.7, 19.3, 15.6.

**17α-Methylpregn-4-ene-3,20-dione (step 2, MQ351).** To a stirred solution of oxalyl chloride (0.174 mL, 2 mmol) in CH₂Cl₂ (15 mL) was added DMSO (0.204 mL, 2.4 mmol) in CH₂Cl₂ (2 mL) at −78 °C. After 10 min, steroid **2** (150 mg, 0.453 mmol) in CH₂Cl₂ (4 mL) was added and stirred for 1 h at −78 °C. Et₃N (0.42 mL, 3 mmol) was added at −78 °C and the reaction was warmed up to room temperature for 1 h. Water was added and the product was extracted into CH₂Cl₂ (100 mL × 2). The solvent was removed, and the residue was dissolved in methanol (10 mL) and then added 3 N HCl (10 mL) and stirred for 30 min. The mixture was extracted with dichloromethane (100 mL × 2). The combined extracts were dried over anhydrous Na₂SO₄, filtered, solvent removed and the residue was purified by flash column chromatography (silica gel, eluted with 10–25% EtOAc in hexanes) to give **MQ351** (100 mg, 67%): ¹H NMR (400 MHz, CDCl₃) δ 5.74 (s, 1H), 2.68–0.92 (m, 19H), 2.13 (s, 3H), 1.20 (s, 3H), 1.14 (s, 3H), 0.92 (s, 3H); ¹³C NMR (100 MHz, CDCl₃) δ 212.2, 199.4, 171.0, 123.7, 61.3, 53.2, 50.4, 44.2, 38.4, 35.6, 35.5, 33.8, 32.7, 32.6, 31.8, 30.9, 27.7, 23.6, 21.4, 20.6, 17.2, 15.7. HRMS (m/z): [M+H]+ calcd. for C₂₀H₂₇NO, 298.2164; found, 298.2164.

e) Procedures for the synthesis of YX54 in six steps.

**17β-Hydroxyandrost-5-en-3-one, 3-cyclic 1,2-ethanediyl acetal (step 1).** To a stirred solution of testosterone (**1**, 2.5 g, 8.68 mmol) in toluene (150 mL) was added ethylene glycol (5 mL) and PTSA (100 mg) at 23 °C. The reaction was refluxed in a flask equipped with a Dean–Stark apparatus for 16 h. After cooling, solid NaHCO₃ (400 mg) was added and stirring continued for 30 min. Water was added and the product was extracted into EtOAc (350 mL). The combined extracts were washed with brine (3 x 100 mL), dried over anhydrous Na₂SO₄, the solvent removed and the residue was purified by flash column chromatography (silica gel, eluted with 25 % EtOAc in hexanes) to afford steroid **2** (2.75 g, 95%): ¹H NMR (400 MHz, CDCl₃) δ 5.33–5.32 (m, 1H), 3.97–3.89 (m, 4H), 3.62 (t, $J$ = 8.6 Hz, 1H), 2.54–2.53 (m, 1H), 2.12–0.92 (m, 19H), 1.02 (s, 3H), 0.74 (s, 3H); ¹³C NMR (100 MHz, CDCl₃) δ 140.1, 121.8, 109.3, 81.7, 64.3, 64.1, 51.2, 49.7, 42.6, 41.7, 36.6, 36.2, 31.8, 31.2, 30.9, 30.3, 23.4, 20.5, 18.8, 11.0, 10.9.

**Androst-5-ene-3,17-dione, 3-cyclic 1,2-ethanediyl acetal (step 2).** To a stirred solution of steroid **2** (2.75 g, 8.4 mmol) in $CH_2Cl_2$ (100 mL) was added $NaHCO_3$ (3.5 g) and Dess−Martin periodinane (5.4 g, 12.7 mmol) at 23 °C. After 1 h, water was added, and the product was extracted into $CH_2Cl_2$ (2 × 100 mL). The combined extracts were washed with brine (2 × 50 mL), dried over anhydrous $Na_2SO_4$, the solvent removed and the residue was purified by flash column chromatography (silica gel, eluted with 15% EtOAc in hexanes) to afford steroid **3** (2.5 g, 89%): $^1$H NMR (400 MHz, CDCl$_3$) δ 5.32–5.30 (m, 1H), 3.94–3.84 (m, 4H), 2.53–2.35 (m, 2H), 2.09–1.03 (m, 17H), 0.99 (s, 3H), 0.82 (s, 3H); $^{13}$C NMR (100 MHz, CDCl$_3$) δ 220.8, 140.1, 121.2, 109.0, 64.2, 64.0, 51.5, 49.6, 47.3, 41.6, 36.5, 36.0, 35.6, 31.2, 31.1, 30.8, 30.4, 21.7, 20.1, 18.7, 13.3.

**17-Trifluoromethanesulfonyl-androsta-5,16-dien-3-one, cyclic 1,2-ethanediyl acetal (step 3).** To a solution of steroid **4** (2.5 g, 7.6 mmol) in THF (80 mL) was added potassium bis(trimethylsilyl) amide (0.5 M in toluene, 25 mL, 12.5 mmol) and N-phenyl-bis(trifluoromethanesulfonimide) (5 g, 14 mmol) at −78 °C. After 1 h, the reaction was allowed to slowly warm to 23 °C. Water was added water and the product was extracted into EtOAc (150 mL × 3). The combined extracts were dried over anhydrous $Na_2SO_4$, the solvent removed and the residue was purified by flash column chromatography (silica gel, eluted with 10% EtOAc in hexanes) to give steroid **4** (3.39 g, 97%): $^1$H NMR (400 MHz, CDCl$_3$) δ 5.56 (s, 1H), 5.34–5.33 (m, 1H), 3.98–3.89 (m, 4H), 2.57–2.54 (m, 1H), 2.24–1.16 (m, 16H), 1.04 (s, 3H), 0.98 (s, 3H); $^{13}$C NMR (100 MHz, CDCl$_3$) δ 159.2, 140.6, 121.3, 114.5, 109.3, 64.4, 64.2, 64.1, 54.2, 50.0, 44.6, 41.7, 36.8, 36.1, 31.7, 30.9, 30.4, 29.9, 28.6, 20.1, 18.7, 15.1.

**3-Oxo-androsta-5,16-diene-17-carbonitrile, cyclic 1,2-ethanediyl acetal (step 4).** To a solution of the steroid **4** (3.39 g, 7.4 mmol) was added Cu (I) I (100 mg), NaCN (700 mg) and tetrakis(triphenylphosphine)palladium (260 mg) under $N_2$. The reaction was refluxed for 2 h and cooled to 23 °C. Aqueous $NaHCO_3$ (40 mL) and water (100 mL) were added and the product was extracted into EtOAc (100 mL × 3). The combined extracts were washed with brine (100 mL), dried over anhydrous $Na_2SO_4$, the solvent removed and the residue was purified by flash column chromatography (silica gel, eluted with 10% EtOAc in hexanes) to give steroid **5** (2.01 g, 80%): $^1$H NMR (400 MHz, CDCl$_3$) δ 6.62 (s, 1H), 5.35–5.34 (m, 1H), 4.00–3.90 (m, 4H), 2.58–2.55 (m, 1H), 2.39–1.06 (m, 16H), 1.06 (s, 3H), 0.93 (s, 3H); $^{13}$C NMR (100 MHz, CDCl$_3$) δ 147.4, 140.5, 127.2, 121.2, 115.8, 109.2, 64.4, 64.1, 55.8, 49.8, 48.0, 41.7, 36.7, 36.1, 33.9, 32.9, 31.1, 30.9, 30.3, 20.4, 18.7, 16.0.

**3-Oxo-androst-5-ene-17β-carbonitrile, cyclic 1,2-ethanediyl acetal (step 5).** To a solution of steroid **5** (2.01 g, 5.9 mmol) in EtOAc (150 mL) was added Pd/C (10%, 100 mg) in a Parr hydrogenation flask. The flask was evacuated and refilled with $H_2$ three times. Hydrogenation was carried out at 55 Psi overnight. The mixture was filtered through celite and washed with EtOAc (100 mL). The solvent was removed and the residue was purified by flash column chromatography (silica gel, eluted with 10% EtOAc in hexanes) to give steroid **6** (2 g, 100%): $^1$H NMR (400 MHz, CDCl$_3$) δ 5.34 (s, 1H), 3.96–3.94 (m, 4H), 2.54–0.97 (m, 20H), 1.04 (s, 3H), 0.93 (s, 3H); $^{13}$C NMR (100 MHz, CDCl$_3$) δ 140.2, 121.4, 121.3, 109.2, 64.4, 64.2, 54.5, 49.3, 44.1, 41.7, 40.0, 36.9, 36.6, 36.2, 32.2, 31.5, 30.9, 26.5, 24.6, 20.6, 18.8, 14.1.

**3-Oxo-androst-4-ene-17β-carbonitrile (step 6, YX54).** To a stirred solution of steroid **6** (2 g, 5.9 mmol) in acetone (100 mL) was added p-toluenesulfonic acid (400 mg) at 23 °C for 16 h. Solid $NaHCO_3$ (1 g) was added and stirred for 15 min. Most of the acetone was removed under reduced pressure. Water was added and the product was extracted into EtOAc (200 mL). The EtOAc was dried over anhydrous $Na_2SO_4$, the solvent removed and the residue was purified by flash column chromatography (silica gel, eluted with 20% EtOAc in hexanes) to give **MQ351** (**7**, 1.62 g, 92%): $^1$H NMR (400 MHz, CDCl$_3$) δ 5.69 (s, 1H),

2.38–0.99 (m, 20H), 1.17 (s, 3H), 0.93 (s, 3H); $^{13}$C NMR (100 MHz, CDCl$_3$) δ 199.2, 170.3, 123.9, 120.9, 53.5, 53.2, 44.0, 39.9, 38.4, 36.6, 35.7, 35.5, 33.7, 32.4, 31.6, 26.3, 24.3, 20.5, 17.2, 14.1. HRMS (m/z): [M+H]+ calcd. for $C_{22}H_{32}O_2$, 329.2475; found, 329.2474.

## Reporting summary

Further information on research design is available in the Nature Portfolio Reporting Summary linked to this article.

## Data availability

All data supporting the findings of this study are available within the manuscript and the supplementary information. Source data are provided with this paper. The data used in the analyses are available to any researchers for the purpose of reproducing or extending the analysis. The cryo-EM density maps and corresponding atomic coordinates of human Kir7.1 have been deposited in the Electron Microscopy Data Bank (EMDB) and the Protein Data Bank (PDB) under accession codes: PIP$_2$-bound extended Kir7.1 PDB: 9PR5; EMD-71798. PIP$_2$-bound docked Kir7.1 9PR6; EMD-71799, and ENT-17OHPC & diC8-PIP$_2$ bound Kir7.1 9PR7; EMD-71800. Previously published PDB codes are: Kir2.1[33] 7ZDZ, Kir2.2[20] 3JYC, Kir3.2[21] 6XIS, and Kir6.2[22] 6C3P. Source data are provided with this paper.

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

## Acknowledgements

We thank Brock Summers, Katherine Basore, and Bradley Readnour at Washington University in St. Louis Center for Cellular Imaging (WUCCI), Kunpeng Li at Case Western Research University (CWRU), Reja-ul Hoq and Min Su at University of Missouri-Columbia (MIZZOU), and Shubhangi Agarwal at the National Center for CryoEM Access and Training (NCCAT) for their assistance with grid screening and data collection. We thank Dr. Sunjoo Lee for helping with the purification protocol, and Citlalli Vergara for helping with cell culture. This work was supported by BJC Investigator fund to P.V.L. We thank Alexandra Shabliy for the initial help with Kir7.1 tagged constructs.

## Author contributions

Q.N. and P.V.L. contributed to the conception and design of this research, data acquisition, and processing and wrote the manuscript. Q.N. was responsible for the entire construct design, cloning and expression, protein purification, grid preparation, data collection, and model building. Q.N. and P.V.L. obtained all electrophysiological data. P.V.L. supported the project since its conception. P.V.L., R.Z., and Z.F. provided day-to-day supervision of the project and helped with data analysis. S.V. helped with data processing. A.R. helped generate initial construct. W.H. assisted with the data collection. J.Y. and J.Z. helped with purification troubleshooting. D.F.C., Y.X., and M.Q. synthesized steroid compounds and performed structure-function analysis. All authors performed/contributed to experiments, analyzed data, contributed to writing, reviewed and approved the final version of the manuscript.

## Competing interests

P.V.L., Q.N., Y.X., M.Q., and D.F.C. are listed on a provisional patent application (63/650,605) related to this study filed by WashU School of Medicine. It has currently entered the PCT stage (WO2025245325A1). The remaining authors declare no competing interests.
