## [Transparent Peer Review file · Nature Communications]

Bioactive Lipid-Mediated Structural and Functional Regulation of the Essential Human Potassium Channel Kir7.1

Corresponding Author: Professor Polina Lishko

Version 0:

Reviewer comments:

Reviewer #1

(Remarks to the Author)

In this paper Niu et al. present for the first time three different cryo-EM structures of the human Kir7.1 in different conformations states

- a) An Extended conformation (E-state) of Kir7.1 complexed with endogenous PIP2 and Cholesterol, which represents a non-conductive state.
- b) Docked conformation is obtained with the channel 7.1 bound only to the endogenous PIP2 This state corresponds to a semi-open configuration
- c) Kir2.1 complexed with a diC8-PIP2 and a synthetic steroid (ENT-17OPC), the most potent activator of Kir7.1, which binds very close to the cholesterol binding pocket. Its conformation resembled the E-state with a slightly enlarged G-loop gate. The hypothesis is that the true open conformation on the channel represents an intermediate state between the D- and E-states

Functional studies (electrophysiology) shows that

- Cholesterol inhibits Kir7.1 channel
- Progesterone (P4) can activate the channel even in absence of PIP2. And there is a cooperativity between PIP2 and progesterone
- Kir7.1 is activated by synthetic steroids particularly ENT-17OPHC (a certain number of endogenous and synthetic steroids long with their stereoisomers were synthesized and tested)

Various points need to be improved and discussed:

- a) The structure of Kir7.1 in complex with the endogeneous PIP2 (activator) shows a docked state and the structure of Kir7.1 in complex with PIP2 and cholesterol (described as an inhibitor) does show an extended non conductive conformation. The channel length is extended from 105 Å (D-state) to 115Å (E-state).

The differences between the apo and PIP2-bound structures has been studied in various Kir channels. In the apo-Kir2.2 crystal structure, the protein is observed in an extended (E-state) conformation and in the PIP2-bound structures, the CTD approaches the membrane's inner surface, translating toward the TMD by 6 Å. (Hansen et al Nature 2011; Tao et al, Science 2009). Similar results about PIP2-induced channel compaction were observed in Kir3.2 cryo-EM structures (Niu et al, eLife, 2020). Apo Kir2.1 was observed in the extended, non-conductive state (Fernandes et al, Sciences Advances 2022). In another hand, the crystal structures of Kir2.2 (5KUM) and Kir3.2 (3SYO and 3SYP) have been obtained in the compact form without PIP2. The compact structure (in presence of PIP2) has also been associated with a gating conformational change (Zangerl-Plessl, J. Gen Physiol2020).

This paper shows for the first time a Kir channel in an extended state when complexed with PIP2. The most plausible explanation for this extended conformation in presence of PIP2 (non conductive) is that the presence of the cholesterol maintains and stabilizes the CTD in the E-state.

Electrophysiological studies shows clearly that the presence of cholesterol inhibits the channel, which is in agreement with this extended state.

The mechanistic basis of the functional effect of the cholesterol remains poorly defined

In a recent work, Barbera et al. (iScience 2022) proposes that cholesterol causes a “decoupling” effect between specific domains within the channel located at the interface between channel subunits. It would be very interesting to compare in details the docked state and the expended state at the level of the interface between subunits and note if there is any

structural differences.

b) In the structure of Kir7.2, the met 125 replaces the highly conserved arginine found in other Kir channels; this substitution explains the permeability to cesium ions. Could this structural distinction explain why this channel is a weak rectifier?

c) The structure of Kir2.1 bound to diC8-PIP2 and the activating steroids, ENT-17OPC surprisingly shows an extended state that is usually associated with a non-conducting state. Although the electrophysiological functional studies clearly show an activation of the Kir7.1 channel. No real explanation is given.

Several approaches could be explored

- The detergent used for the extraction, purification and cryo-EM studies of the Kir7.1 is the GDN glycol-diosgenin. This is a steroid-derived detergent. It cannot be ruled out that GDN may compete with other steroids for interaction with sterol-binding sites on the channel. It has been reported that GDN binding could affect the conformation of protein (see Di Trani et al. doi: 10.1073/pnas.2205228119 "Structural basis of mammalian Complex IV inhibition by steroids"). Competition for the steroid binding site between cholesterol or ENT-17OPC and GDN glycol-diosgenin should be discussed and could explain the surprising results obtained with ENT-17OPC (conformational E-state).

-The final hypothesis is that the true open conformation on the channel represents an intermediate state between the D- and E- states

Observation of the workflow for the calculation of the cryo-EM structures, reveals that the final structures were calculated after the elimination of a large number of particles. This demonstrates the dynamics of the protein in solution, and the presence of various populations of different conformational states. During the image analysis process, many 3D classes are eliminated but their exploration could provide and reveal information on various conformations present in the same sample. The proposed model, intermediate between D and E states could be present in solution but in a more restricted population (this was described in Fernandes et al. *Science Advances* 2022)

d) Of the three structures shown, only one (extended state) shows the presence of two potassium ions in the selectivity filter. Surprisingly the docked conformation (more conductive state) does not show any potassium ions. Is there any explanation?

Reviewer #2

(Remarks to the Author)

The paper presented by Niu, Vu and co-authors describes structural and functional data on Kir7.1 channels. These channels are known to play key physiological roles, in the retina particularly, yet have poorly developed pharmacology and key understanding of their modulation remains unknown. The current manuscript therefore is important as it advances both areas simultaneously. Previously the authors identified endogenous and synthetic ligands which can promote channel activity. The mechanism of these small molecules was not known and this weakness hindered further development of higher-affinity Kir7.1 modulators. The authors now present new high-resolution structural data on a variety of physiologically relevant ion channel conformations. These include the structures at 2.8–4.0 Å resolution in multiple functional states. Notably these include a PIP₂-bound extended "E"-state and PIP₂-bound docked "D"-state. An agonist-bound state is also described. A major finding is the large conformational change that is found between the E and D states but overall these structural data are considered to be a significant development in identifying a framework for function and pharmacology of Kir7.1 channels. These data highlight a number of key insights regarding the basic function of Kir7.1 channels. For example, the authors find significant structural difference within the selectivity filter owing to a natural amino acid variation (M125) that appears to disrupt a salt-bridge that is found in other Kir channels. This change likely allows for the permeation in Cs⁺ Kir7.1 channels but not in other isoforms.

Overall the manuscript is well written for and accurately referenced. I believe the work is of very high impact and adds significantly to the understanding of Kir7.1 channels as well as charting new territory into novel pharmacology.

Comments:

While the structural data are very well described and presented, I have some comments regarding the electrophysiology.

1) Dose-response relationships are shown for ENT-P4 and ENT-17HPC (Figure 5) but are lacking for the other compounds. For instance, in Fig 4f, the work is being performed at the steepest phase of the relationship, where 10 μM P4 has essentially no effect while 30 μM P4 potently stimulates current. Are there dose-response relationships available to reference the concentrations used in these studies?

2) The experimental layout as representative traces are difficult to follow. The methods report that these data are obtained with whole-cell voltage-clamp. Does this mean that all of the agents are membrane permeant or are that added to the internal solution prior to going whole cell? Or is this a mix of impermeant in the internal and permeant that is subsequently added to the bath. Perhaps a schematic of the experiment in combination with a diary plot showing the current density would provide a visual improvement for the reader.

3) Are there wash-out controls for after the currents have been maximally activated?

Minor:

Figure 7 e. the "putative conductive model" shown in light blue (far right) appears to show a non-conductive selectivity filter. Can the authors clarify if this is truly different from the docked model (pink) or if this is just variability in the cartoons?

Version 1:

Reviewer comments:

Reviewer #1

(Remarks to the Author)

I have read the new manuscript, major changes have been made, new experiments have been conducted, and clarifications have been provided, particularly in the discussion, figures were added.

With these modifications, I believe that this work will have an important impact and will contribute significantly to the understanding of the gating mechanism and modulation of Kir 7.1 channels and other channels and will open up new perspectives in pharmacology.

Reviewer #2

(Remarks to the Author)

The authors have addressed all concerns and comments. I have no further suggestions and congratulate them on their excellent study.

Re: Response to Reviewers Comments/ manuscript NCOMMS-25-57549-T

We thank the Reviewers for their constructive and helpful feedback, as well as for their time and efforts. We have revised the manuscript according to the suggestions. Below is our point-by-point response (in italics). Please note that the figure order and references have been updated to reflect the requested changes. The edited parts of the manuscript and references are highlighted in yellow and summarized below:

1. Updated Figure 2c to illustrate the curvature difference of the outer helix in the docked and extended conformations. The previous panel 2c has been moved to Supplementary Figure S7.
2. Updated Figure 3e and 3f to include cartoons illustrating the patch-clamp conditions for improving clarity.
3. Performed additional experiments and added a new panel, Figure 4g, illustrating dose-response for progesterone and its competition by cholesterol. The previous panel Figure 4f has been moved to Figure S8, and previous Figure 4g has been renumbered as Figure 4f.
4. Added a new panel, Figure 5a, using a schematic cartoon to clarify the electrophysiology experimental conditions, including the pipette solution, bath solution, and the application of the compounds.
5. Created a new Figure 8 by separating the previous Figure 7e into an independent figure. Created new Figure 9.
6. Added a new Supplementary Figure S7b to show that steroid ligand binding can alter the curvature of the outer helix.

REVIEWER COMMENTS:

Reviewer #1 (Remarks to the Author):

...a) ...The differences between the apo and PIP2-bound structures has been studied in various Kir channels. In the apo-Kir2.2 crystal structure, the protein is observed in an extended (E-state) conformation and in the PIP2-bound structures, the CTD approaches the membrane's inner surface, translating toward the TMD by 6 Å. (Hansen et al Nature 2011; Tao et al, Science 2009). Similar results about PIP2-induced channel compaction were observed in Kir3.2 cryo-EM structures (Niu et al, eLife, 2020). Apo Kir2.1 was observed in the extended, non-conductive state (Fernandes et al, Sciences Advances 2022). In another hand, the crystal structures of Kir2.2 (5KUM) and Kir3.2 (3SYO and 3SYP) have been obtained in the compact form without PIP2. The compact structure (in presence of PIP2) has also been associated with a gating conformational change (Zangerl-Plessl, J. Gen Physiol2020).

A: We thank the reviewer for these in-depth comments and have referenced all of these suggested papers in the revised manuscript.

b)...This paper shows for the first time a Kir channel in an extended state when complexed with PIP2. The most plausible explanation for this extended conformation in presence of PIP2 (non conductive) is that the presence of the cholesterol maintains and stabilizes the CTD in the E-state. Electrophysiological studies shows clearly that the presence of cholesterol inhibits the channel, which is in agreement with this extended state. The mechanistic basis of the functional effect of the cholesterol remains poorly defined In a recent work, Barbera et al. (iScience 2022) proposes that cholesterol causes a “decoupling” effect between specific domains within the channel located at the interface between channel subunits. It would be very interesting to compare in details the docked state and the expended state at the level of the interface between subunits and note if there

is any structural differences.

A: We agree that this study represents the first report of a Kir channel maintaining an extended conformation in the presence of PIP₂, and we followed reviewer's recommendations and performed in depth comparison between two states. We found that cholesterol binding increases the rigidity of the transmembrane (TM) helices, preventing them from turning inward and thereby restricting the movement required for CTD lifting and compaction. When PIP₂ is present and cholesterol is absent from the steroid binding pocket (SBP), the TM helices can move more freely. Only when the helices compress to an appropriate degree is sufficient space created for CTD rotation and docking. In contrast, cholesterol binding increases TM rigidity and effectively locks the helices at a fixed angle, preventing this conformational transition. Therefore, we propose that cholesterol removal is essential for enabling the transition between the extended and docked states. We have also added a new panel in Supplementary Figure S7, updated Figure 2c, and clarified in the manuscript the curvature change of the outer helix upon steroid binding at the SBP. In the absence of bound steroids, the pocket remains flexible, allowing the outer helix to bend at a larger angle. In contrast, cholesterol binding straightens the outer helix, consistent with its stabilizing effect on the extended conformation.

c)...In the structure of Kir7.2, the met 125 replaces the highly conserved arginine found in other Kir channels; this substitution explains the permeability to cesium ions. Could this structural distinction explain why this channel is a weak rectifier?

A: We thank the reviewer for raising this important comparison. We believe that the Met125 substitution primarily contributes to the Cs⁺ selectivity rather than to rectification strength. Among the Kir family, Kir2.1 exhibits the strongest rectification, Kir3.2 shows intermediate rectification, and Kir7.1 is the weakest. This gradient can be directly explained by differences at three key structural sites: the inner helix cavity, the G-loop gate, and the CTD entrance. We added the following explanation in the Discussion section:

“...At the level of first checkpoint in Kir2.1, i.e. CTD entrance, D255 constricts the passage to ~5.6 Å, entrapping hydrated Mg²⁺ (~8 Å) and preventing K⁺ permeability (Fig. 9, purple). If partially hydrated Mg²⁺ (~4.8 Å) passes this gate, it will be further entrapped at the next checkpoint, i.e. the G-loop gate, which is formed by Glu224 and Glu299. These glutamates introduce additional negative charges by lining up 9.7 Å pore and therefore stabilizing Mg²⁺ within. The final checkpoint, i.e. inner helix, contains Asp172 that provides additional electrostatic anchor for Mg²⁺. Together, these acidic anchors and tight geometry explain why Kir2.1 nearly abolishes outward K⁺ flux at depolarized voltages³³.

Similar situation exists in Kir3.2 (GIRK2), with few exceptions²¹. At the CTD entrance, instead of electrostatic trap, Tyr266 and Tyr267 form ~7.5 Å hydrophobic constriction (Fig. 9, orange). The second checkpoint is formed by Phe192, a hydrophobic residue that does not stabilize cations. Finally, at G-loop, the narrowest site is defined by Met313, forming a much wider ~14.6 Å opening, which cannot trap hydrated Mg²⁺. However, acidic residues in the cytoplasmic vestibule (e.g., Glu236) still provide some stabilization. As a result, Kir3.2 supports inward rectification, but the block is weaker and more transient than in Kir2.1.

However, most acidic anchors outlined above are absent in Kir7.1 (Fig. 9, grey). At the CTD entrance, Asp232 forms a relatively wide ~12.5 Å passage, too wide to effectively coordinate Mg²⁺. The second checkpoint, the G-loop, is similar in size to Kir2.1 (~9.4 Å), however it formed by nonpolar Ile281, thus, lacking any charge stabilization. The final checkpoint at the inner helix gate is represented by another nonpolar residue, Val157. Thus, Kir7.1 allows outward K⁺ currents with little suppression, producing weakest rectification. In summary, the progressive loss of acidic residues in addition to widening at key checkpoints explains the diversity of rectifications.”

Additionally, we have added new Figure 9.

d).. The structure of Kir2.1 bound to diC8-PIP2 and the activating steroids, ENT-17OPC surprisingly shows an extended state that is usually associated with a non-conducting state. Although the electrophysiological functional studies clearly show an activation of the Kir7.1 channel. No real explanation is given. Several approaches could be explored

- The detergent used for the extraction, purification and cryo-EM studies of the Kir7.1 is the GDN glycol-diosgenin. This is a steroid-derived detergent. It cannot be ruled out that GDN may compete with other steroids for interaction with sterol-binding sites on the channel. It has been reported that GDN binding could affect the conformation of protein (see Di Trani et al. doi: 10.1073/pnas.2205228119 "Structural basis of mammalian Complex IV inhibition by steroids"). Competition for the steroid binding site between cholesterol or ENT-17OPC and GDN glycol-diosgenin should be discussed and could explain the surprising results obtained with ENT-17OPC (conformational E-state).

A: We thank the reviewer for this insightful comment and for highlighting the potential influence of detergent effects on the observed conformation. In our experiments, the same detergent was used for all structural samples, including the docked state without bound steroids, in which both the ENT-17OHPC-bound and cholesterol-bound structures exhibited the relatively extended conformation. This observation suggests that the extended state is likely ligand-induced rather than detergent-driven. Nevertheless, we agree that the use of GDN could in principle influence the conformational equilibrium of Kir7.1 by partially occupying or competing for the steroid-binding pocket, which may be one of the factors preventing visualization of a fully open conformation in the ENT-17OHPC-bound structure.

To address this point further, we have expanded the discussion section to discuss possible mechanistic scenarios. One possibility is that multiple ligand molecules act cooperatively to stabilize the open state, and that only long-chain PIP₂ provides sufficient anchoring strength to pull the CTD upward. We also acknowledge that GDN may compete with cholesterol or ENT-17OHPC at some of the four potential binding sites, which could hinder stabilization of a fully open channel. Together, these considerations have been incorporated into the revised Discussion, where we describe the interplay among ligand stoichiometry, cooperative binding, and detergent competition as plausible explanations for the observed extended conformation in the ENT-17OHPC-bound structure.

Specifically, we added the following part in the Discussion: "We propose that cholesterol binding increases the rigidity of the transmembrane helices, preventing them from turning inward and thereby restricting the movement required for CTD lifting and compaction (Fig. S7b). In contrast, when PIP₂ is present and the steroid binding pocket (SBP) is unoccupied, the TM helices can move more freely. Only when the helices compress to an appropriate degree is sufficient space created for CTD rotation and docking. Thus, cholesterol binding effectively locks the helices at a fixed angle, straightening the outer helix and stabilizing the extended conformation by preventing the "kinked" flexibility that normally facilitates CTD docking. Consistent with this interpretation, structural analysis shows that in the absence of bound steroids, the SBP remains flexible, allowing the outer helix to bend at a larger angle.

Therefore, cholesterol removal could be essential for enabling the conformational transition between the extended and docked states. However, in the context of a living cell, complete cholesterol sequestration is impossible, unless it is replaced by other steroid-like entity. Hence, we propose that a replacement of cholesterol with activating steroids, such as progesterone, pregnenolone, or ENT-17OHPC, facilitates a transition to an open state. By occupying SBP, these activating steroids would reduce the rigidity of the transmembrane helices observed in the E-state, and allow intermediate CTD rotation (Fig. 8). Such intermediate rotation should open both gates and produce a conductive state of the channel as illustrated on Fig. 8 (blue model). We suspect that a conformation, currently represented by ENT-17OHPC-bound state, captures a channel in a primed but not yet conductive state. This primed conformation could result from the presence of synthetic PIP₂ used during

purification. By competing with endogenous long-chain PIP₂, synthetic short-chain PIP₂ provides weaker anchoring strength for CTD docking. Another possibility is a presence of GDN detergent, a steroid-derived amphiphile, that may compete with ENT-17OHPC for the same binding pocket. In such scenario, only a portion of four SBP in the homotetramer are occupied by ENT-17OHPC, while other are occupied by GDN, therefore preventing complete ligand occupancy and stabilization of the fully open channel. Further investigation will be necessary to reveal a conductive state of Kir7.1.”

e) ..The final hypothesis is that the true open conformation on the channel represents an intermediate state between the D- and E- states. Observation of the workflow for the calculation of the cryo-EM structures, reveals that the final structures were calculated after the elimination of a large number of particles. This demonstrates the dynamics of the protein in solution, and the presence of various populations of different conformational states. During the image analysis process, many 3D classes are eliminated but their exploration could provide and reveal information on various conformations present in the same sample. The proposed model, intermediate between D and E states could be present in solution but in a more restricted population (this was described in Fernandes et al Sciences advance 2022)

A: We thank the reviewer for the suggestion. We carefully re-analyzed the discarded particles in our dataset. However, these particles appeared to be of low quality, likely due to thick ice or sample damage. Consequently, we were unable to obtain 3D reconstructions at sufficient resolution to draw meaningful conclusions about distinct conformational states.

f) .. Of the three structures shown, only one (extended state) shows the presence of two potassium ions in the selectivity filter. Surprisingly the docked conformation (more conductive state) does not show any potassium ions. Is there any explanation?

A: We thank the reviewer for pointing out this difference. Indeed, we observed weak densities consistent with potassium ions in the selectivity filter of both docked and ENT-17OHPC-bound states. This likely reflects partial occupancy rather than a non-conductive conformation. Since these ion-like densities were not sufficiently strong to allow us confidently place K⁺ ions in the model, we chose not to include K⁺ ions in these structures.

Reviewer #2 (Remarks to the Author):

The paper presented by Niu, Vu and co-authors describes structural and functional data on Kir7.1 channels. These channels are known to play key physiological roles, in the retina particularly, yet have poorly developed pharmacology and key understanding of their modulation remains unknown. The current manuscript therefore is important as advances both areas simultaneously. Previously the authors identified endogenous and synthetic ligands which can promote channel activity. The mechanism of these small molecules was not known and this weakness hindered further development of higher-affinity Kir7.1 modulators. The authors now present new high-resolution structural on a variety of physiologically relevant ion channel conformations. These include the structures at 2.8–4.0 Å resolution in multiple functional states. Notably these include a PIP₂-bound extended “E”-state and PIP₂-bound docked “D”-state. An agonist-bound state is also described. A major finding is the large conformational change that is found between the E and D states but overall these structural are considered to be a significant development in identifying a framework for function and pharmacology of Kir7.1 channels.

These data highlight a number of key insights regarding the basic function of Kir7.1 channels. For example, the authors find significant structural difference within the selectivity filter owing to a natural amino variation (M125) that appears to disrupt a salt-bridge that is found in other Kir channels. This change likely allows for the permeation in Cs⁺ Kir7.1 channels but not in other isoforms.

Overall the manuscript is well written for and accurately referenced. I believe the work is of

very high impact and adds significantly to the understanding of Kir7.1 channels as well as charting new territory into novel pharmacology.

A: We thank the reviewer for the enthusiastic and encouraging comments.

Comments:

While the structural data are very well described and presented, I have some comments regarding the electrophysiology.

1) Dose-response relationships are shown for ENT-P4 and ENT-17HPC (Figure 5) but are lacking for the other compounds. For instance, in Fig 4f, the work is being performed at the steepest phase of the relationship, where 10 μ M P4 has essentially no effect while 30 μ M P4 potently stimulates current. Are there a dose-response relationships available to reference the concentrations used in these studies?

A: We thank the reviewer for this helpful suggestion. We used 10 μ M P4, which corresponds to its EC_{50} concentration as determined by our previous work (Bjorkgren et al., JGP 2021 and Haoui et al, Science Advances 2025). To clarify this point, we have now included dose-response analyses for progesterone alone and for progesterone in the presence of cholesterol, illustrating clear competition between two steroids. Specifically, progesterone was tested at 1 μ M, 10 μ M, 30 μ M, 100 μ M, and 300 μ M, with or without 30 μ M cholesterol. The results show a clear rightward shift of the EC_{50} (from 12 μ M to 26 μ M) when cells were treated with both progesterone and cholesterol, indicating that cholesterol competes with progesterone and reduces its potency. These results have been incorporated into Figure 4g of the revised manuscript.

2) The experimental layout as representative traces are difficult to follow. The methods report that these data are obtained with whole-cell voltage-clamp. Does this mean that all of the agents are membrane permeant or are that added to the internal solution prior to going whole cell? Or is this a mix of impermeant in the internal and permeant that is subsequent added to the bath. Perhaps a schematic of the experiment in combination with a diary plot showing the current density would provide a visual improvement for the reader.

A: We apologize for the confusion. All steroid compounds, including progesterone and its analogs, were initially dissolved in either ethanol or DMSO to prepare high-concentration stock solutions, which were then diluted with bath solutions to the desired concentrations and applied to the cells via perfusion lines during whole-cell patch-clamp recordings as described in the Methods. To improve clarity, we have added schematic cartoons in Figures 3e, 3f, and 5a to illustrate the patch-clamp configuration and the application method used in these experiments.

3) Are there wash-out controls for after the currents have been maximally activated?

A: Yes, for each steroid application, we waited until the current reached a steady-state maximal activation before applying bath solution for washout and/or inhibiting the current with ML418, which is specific inhibitor of human Kir7.1. Each steroid was tested on an independent cell, and no multiple applications of different steroids were performed on the same cell during dose-response experiments.

Minor:

Figure 7 e. the “putative conductive model” shown in light blue (far right) appears to show a non-conductive selectivity filter. Can the authors clarify if this is truly different from the docked model (pink) or if this is just variability in the cartoons?

A: We thank the reviewer for pointing out this difference in the model. We have updated the model in Figure 8 to more accurately represent the conductive state and have provided

additional details and clarification regarding this model in the revised manuscript text.

Re: Response to Referees' Comments/ manuscript NCOMMS-25-57549A:

REVIEWERS' COMMENTS

Reviewer #1 (Remarks to the Author):

I have read the new manuscript, major changes have been made, new experiments have been conducted, and clarifications have been provided, particularly in the discussion, figures were added.

With these modifications, I believe that this work will have an important impact and will contribute significantly to the understanding of the gating mechanism and modulation of Kir 7.1 channels and other channels and will open up new perspectives in pharmacology.

Answer: We thank the Reviewer for their enthusiastic and encouraging comments, as well as for their time and efforts.

Reviewer #2 (Remarks to the Author):

The authors have addressed all concerns and comments. I have no further suggestions and congratulate them on their excellent study.

Answer: We thank the Reviewer for their enthusiastic and encouraging comments, as well as for their time and efforts.